# Knowledge acquisition is governed by striatal prediction errors

Alex Pine[1,2], Noa Sadeh[2], Aya Ben-Yakov[2,3], Yadin Dudai[2] & Avi Mendelsohn [ID] [1,4]

Discrepancies between expectations and outcomes, or prediction errors, are central to trial-and-error learning based on reward and punishment, and their neurobiological basis is well characterized. It is not known, however, whether the same principles apply to declarative memory systems, such as those supporting semantic learning. Here, we demonstrate with fMRI that the brain parametrically encodes the degree to which new factual information violates expectations based on prior knowledge and beliefs—most prominently in the ventral striatum, and cortical regions supporting declarative memory encoding. These semantic prediction errors determine the extent to which information is incorporated into long-term memory, such that learning is superior when incoming information counters strong incorrect recollections, thereby eliciting large prediction errors. Paradoxically, by the same account, strong accurate recollections are more amenable to being supplanted by misinformation, engendering false memories. These findings highlight a commonality in brain mechanisms and computational rules that govern declarative and nondeclarative learning, traditionally deemed dissociable.

---

[1] Sagol Department of Neurobiology, University of Haifa, Haifa 3498838, Israel. [2] Department of Neurobiology, Weizmann Institute of Science, Rehovot 76100, Israel. [3] MRC Cognition and Brain Sciences Unit, University of Cambridge, Cambridge CB27EF, UK. [4] The Institute of Information Processing and Decision Making (IIPDM), University of Haifa, Haifa, Israel. These authors contributed equally: Yadin Dudai, Avi Mendelsohn. Correspondence and requests for materials should be addressed to A.P. (email: alexpine@cantab.net) or to A.M. (email: amendels1@univ.haifa.ac.il)

The brain is remarkably adept at learning from past experiences to make predictions about future states of the world. Computationally, predictions can be optimized by the calculation of an error term associated with their accuracy in light of new information—that is, a mismatch between expectation and reality. These prediction errors (PEs) are used to update the brain's beliefs and models of the world, in order to engender superior future predictions and minimize the error term[1–3]. This principle has been most successfully demonstrated in reinforcement learning, which concerns the acquisition, and updating, of action– or stimulus–outcome associations through cumulative experiences. Learning of this nature, in the face of reward and punishment, has been shown to accord with normative computational models, such as the Rescorla–Wagner model of Pavlovian conditioning[4]. Axiomatic to these models is the generation of outcome-based PEs that update ensuing expectations in proportion to their magnitude. They have also been well documented in the brain[5,6], where it has been posited that the phasic activity of dopaminergic neurons in the ventral tegmental area (VTA) encodes PEs for reward, firing in response to unpredicted reward (positive PE) and pausing in response to unexpected omission of reward (negative PE)[7]. Numerous functional magnetic resonance imaging (fMRI) studies have identified hemodynamic activity correlating with reward PEs, particularly in the ventral striatum (VS), a primary efferent target of VTA neurons[8–11].

Remarkably, despite the acclaim of this computational account, it is unknown whether the same principles apply to the learning of facts (semantic knowledge) and events (episodic memory) which constitute declarative memory. These memories—unlike those pertaining to nondeclarative forms of learning, such as reinforcement learning, skill learning, and other acquired behaviors—enable conscious retrieval of personal experience and knowledge acquired throughout one's lifetime[12], and are traditionally viewed as being neurobiologically dissociable from nondeclarative memories[13–15]. Indeed, it is a commonly held view that a key factor in the acquisition of knowledge is the number of exposures to information (repetitions), yet the relationship between memory and study repetition is not straightforward. In certain cases, rather than enhancing memory, repetition bears no consequence and can even be detrimental to long-term retention[16–18]. Such anomalies would be explicable if prediction-error-based updating is globally applicable to memory, whereby the surprise of new information in light of what we already know and believe should serve as a prominent driving force of learning. Mere repetition would be of limited value because a predictable message is uninformative. This is implicit in Shannon's influential theory[19] which defines information in terms of surprise.

There are a number of existing findings which hint at the latter possibility. First, medial temporal lobe (MTL) structures considered to play a critical role in declarative memory—particularly the hippocampus (HC) and adjacent cortices—seem to be strongly attuned to novelty and mismatch detection. For example, enhanced MTL activity in fMRI is evoked by simple perceptual stimuli deemed surprising, or which violate expectations based on prior statistical regularities/associations, such as a change in the temporal order of stimuli presented in a repeating sequence[20–25]. Thus, it has been posited that the HC generates predictions about how events will unfold—based on past experiences—and detects mismatches between these expectations and events as they occur[21,26]. Behaviorally, surprising episodes are sometimes remembered with greater fidelity in later memory tests[20,27]. A related phenomenon, termed hypercorrection, is found in the error-correction field, whereby errors in tests of general knowledge are more likely to be corrected when they are committed with high confidence[28].

Second, striatal and dopaminergic midbrain structures typically associated with nondeclarative learning are increasingly implicated in declarative memory. For example, in neuroimaging studies, the caudate nucleus is sometimes more active during encoding of subsequently remembered vs. forgotten episodic memoranda[29,30]. Moreover, a burgeoning literature is revealing a causal role for dopamine and reward processes in augmenting declarative memory[31,32]. Thus, anticipation of monetary reward or punishment, and motivation to obtain reward, enhances memory for coincident visual stimuli. This phenomenon is associated with greater responses and functional connectivity in a network involving the MTL, dopaminergic midbrain, and VS, in response to reward-related memoranda[31–34]. Pharmacological intervention in humans, with L-dopa and monoaminergic stimulants which enhance dopamine transmission, has also been shown to improve declarative memory[35–37].

Some of the abovementioned findings have been interpreted within the framework of a prominent theory, delineating a hippocampal–striatal–VTA loop, which regulates the entry of information into memory[26]. According to this putative model, hippocampal novelty/unexpectancy signals are conveyed via the VS to the VTA, where they contribute—along with motivation-related information—to the firing of dopamine neurons. Direct VTA–MTL projections in turn facilitate dopamine release in the HC, enhancing long-term potentiation, providing a mechanism by which salient and informationally rich stimuli can enhance learning. Along these lines, the traditional view of a neurobiological, cortico-hippocampal, and midbrain–basal ganglia dissociation for declarative vs. nondeclarative learning, is being replaced by a more nuanced approach which favors an interaction between memory systems[23,31,38–40]. A PE-based account of declarative learning would go a step further, by implying shared rules and neurobiological substrates between some forms of these seemingly disparate memories. The latter approach conforms to process-based memory categorization, which distinguishes different forms of memory by the type of neural computation they depend on, rather than the involvement of consciousness[38,41]. However, critical empirical evidence toward this account remains elusive because it is unknown as to whether information itself evokes PEs that are tracked by the brain and determine long-term memory formation. The effects of coincident extrinsic reward or reinforcement-derived PEs on memory for pictures are typically weak and do not scale with reward magnitude, and the memoranda themselves do not entail a PE—leaving open the question as to whether prediction-error-based learning is inherent to normal declarative memory.

We addressed this lacuna by asking whether the acquisition and updating of declarative knowledge are governed by Rescorla–Wagner-type rules and can be mathematically modeled accordingly. We hypothesized that new information which counters prior knowledge or beliefs—be they erroneous or accurate—evokes prediction errors that engender superior learning and incorporation of information into memory. This behavioral hypothesis was complemented by a neurobiological investigation, examining whether the brain implements PE-based declarative learning by capitalizing on systems already in place for nondeclarative learning. We predicted that the VS encodes a PE based on discrepancies between new information and existing declarative knowledge, as it does for violations of expected reward and punishment. To test these hypotheses, we designed a naturalistic protocol to study factual learning, in a manner which is typical of the way knowledge is imparted and assessed in educational systems. Feedback to questions probing previously studied materials enabled us to mathematically model declarative PEs, delineate their neural substrates with fMRI, and assess their effects on subsequent memory. Our approach exploited the commonplace tendency to forget or incorrectly remember information, as well as human susceptibility to adopt false

memories. Because declarative recollection, whether accurate or false, is accompanied by varying degrees of perceived memory strength, we could determine how the efficacy of learning from feedback relates to its informational PE, in terms of the magnitude of recollection–outcome mismatches.

We found that the degree of memory updating from feedback was directly proportional to the semantic prediction error the feedback information elicited, both in the case of incorrect recollections and in the adoption of false-feedback information in place of accurate memories. Furthermore, activity in the striatum, frontal, and parietal cortices correlated on a trial-by-trial basis with the feedback-evoked PE values (positive and negative), and was predictive of subsequent memory performance. A separate network correlated with the salience (unsigned PE) of the feedback information.

## Results

**Generating prediction errors for factual information.** In a 3-day study (Fig. 1a; Methods), participants were initially requested to read a detailed text containing information regarding a historical event unfamiliar to them (the Falklands War), and were encouraged to encode this knowledge for a test (Test1) which took place 2 days later.

We performed two separate studies: one addressing behavior and another with fMRI. In the former, participants' memory was probed using cued recall, which required them to answer questions concerning information they had read in the text during the study phase. Following each question, participants

were requested to record their confidence in their answers by rating from 0 to 100 their subjective degree of certainty that they had answered correctly. The correct answer to the question was then supplied in the form of a quotation from the original text (the feedback phase). In a subset of trials, novel (i.e., false) answers were displayed during the feedback phase (Fig. 1b). Participants subsequently returned for a second (surprise) test 1 week later (Test2), where they were asked the same set of questions and again rated their confidence (Fig. 1a). The fMRI study employed recognition-memory questions, and Test1 was carried out in the scanner (Fig. 1c).

Our primary question was whether learning and updating from the feedback provided in Test1 was more effective when it conflicted with prior expectations (erroneous or veridical) and accordingly, whether the magnitude of this PE during encoding directly impacted long-term subsequent memory performance. The key to this was mathematically determining the PE (outcome–expectancy) for each trial of learning, which was afforded by the confidence measure—a proxy of memory strength, and therefore of expectancy concerning the information conveyed by feedback (Methods).

**PE magnitude in Test1 correlates with subsequent memory.** We initially analyzed memory performance in Test2 for questions which were answered incorrectly in Test1. In the recall study, an average of 64% (±2.3%) of questions were answered erroneously in Test1, of which 26% (±2.0%) were subsequently answered correctly in Test2. As expected, performance in the recognition

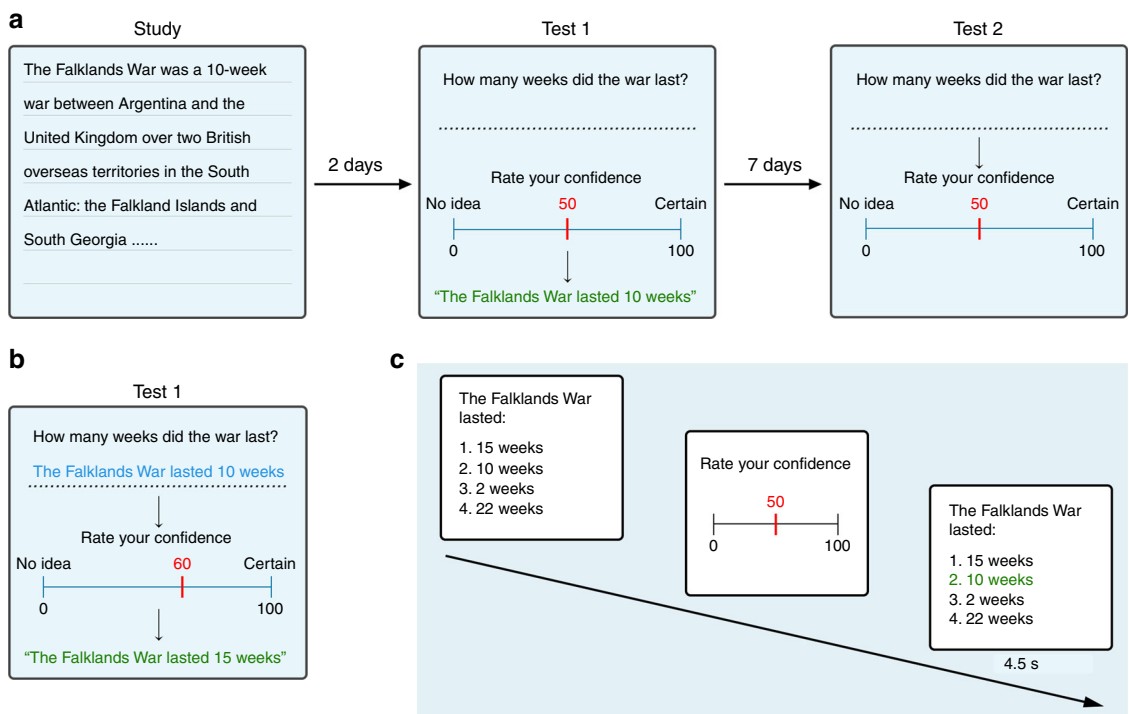

**Fig. 1** Experimental paradigms of declarative knowledge acquisition and updating. **a** The experiments commenced with the study of a lengthy, factual text concerning an unfamiliar historical event. Participants were required to recall information from the text in a test 2 days later and state their confidence in each answer. Feedback (depicted in green) providing the correct answer (outcome) elicited PEs which were hypothesized to depend on the confidence ratings (expectation), such that an erroneous answer expressed with high confidence would engender a large negative PE and vice versa. For each trial, a PE term was calculated as the additive inverse of the confidence (–confidence) for incorrect answers, and 100-confidence for correct answers (positive PEs). An identical (unexpected) test, 1 week later, assessed the degree of learning and memory updating from feedback in Test1. **b** In a subset of trials in the recall study, a novel (false) answer was presented as feedback in order to evoke (negative) PEs for correctly answered questions (the false-memory condition). Test2 answers determined in which trials originally correct memories were supplanted by false information. **c** The paradigm was modified for an fMRI study, where Test1 was carried out in the scanner to examine neural responses to semantic PEs during feedback. Since typing is impractical in the scanner, testing took the form of multiple-choice (recognition) questions, requiring participants to select one out of four potential answers

(fMRI) study was superior, with 41% (±2.4%) of questions answered incorrectly in Test1, of which 48.8% (±3.3%) were subsequently answered correctly in Test2 (Supplementary Figure 1). Due to the feedback and the relatively short intertest period, questions answered correctly in Test1 were seldom answered incorrectly in Test2 (11.9 ± 1.6% in recall and 8.6 ± 0.7% in recognition), precluding meaningful behavioral analysis of the relationship between positive PEs and subsequent memory.

Incorrectly answered questions from the recall experiment were grouped into four bins of prediction error for each participant, and a subsequent accuracy score was calculated for each bin (Methods). This score revealed that the degree to which participants learned from the feedback and updated their knowledge was directly dependent upon the PE magnitude arising from the feedback (main effect of PE; $F_{(3,57)} = 19.8$, $p < 0.001$). Thus, on average, Test2 accuracy was nearly three times greater for questions in which participants were highly confident that their incorrect answer was correct in Test1 (large negative PE), relative to those where they did not supply any response or rated zero confidence in the veracity of their answer (Fig. 2a). Post hoc $t$-tests (Bonferroni corrected) revealed that high PE subsequent accuracy (mean = 55.4 ± 6.6%) was significantly greater than that of the other levels (0 mean = 20.7 ± 2.1%, $t_{(19)}$ = 5.7, and $p < 0.001$; low mean = 24.4 ± 2.1%, $t_{(19)}$ = 4.9, and $p < 0.001$; and medium mean = 30.4 ± 3.5%, $t_{(19)}$ = 3.9, and $p < 0.01$), as well as a difference between medium and 0 PE accuracy of borderline significance ($t_{(19)} = 2.8$, $p = 0.076$).

Results of the fMRI study recognition tests similarly demonstrated a significant correlation between PE and updating for incorrectly answered questions (Fig. 2c). This data set had a smoother distribution of PEs (Supplementary Figure 2; Methods), so we modeled Test2 accuracy with linear regressions, using PE as a predictor. A positive and significant relationship between the PE and subsequent accuracy was found across subjects (mean $\beta$ = 0.19 ± 0.08, $t_{(26)}$ = 2.22, and $p < 0.05$). A regression analysis of the group median subsequent accuracy (Supplementary Figure 3) revealed that each unit increase in magnitude of the negative PE, led to a corresponding increase in Test2 accuracy of roughly half a percent ($\beta = 0.55$, intercept = 29.2, $R^2 = 0.63$, and $p < 0.005$). Thus, median subsequent accuracy for 0 PE trials was 38.5%, as opposed to 100% for the −100 PE trials.

**PE determines adoption of false feedback in place of correct memory.** The false-memory condition was incorporated to engender the occurrence of PEs upon viewing false feedback to correctly answered questions (stated with a confidence of 70 or less, to preclude awareness). We posited that deceiving participants into believing that they had answered erroneously would give rise to negative PEs, again estimated numerically as the inverse of the confidence placed in their (correct) answer. These trials enabled us to test the proposition that stronger accurate memories should be more amenable to being supplanted by false memories, because they should elicit greater PEs upon provision of the fictitious feedback.

On average, 29.6% (±3.6%) of correctly answered questions were met with false feedback. These trials were divided into low and medium PE bins of roughly equal size. We compared the proportion of trials where participants subsequently supplied the false-feedback information in their Test2 answers (rather than the original correct answer, or an incorrect answer—Supplementary Figure 5). In accordance with the hypothesis, medium PE trials (correct, medium confidence) were significantly more likely to lead to the adoption and subsequent recollection of the false feedback than low PE trials (correct, low confidence) (Fig. 2a; mean percent false-feedback answers = 32 ± 5.8% vs. 51.6 ± 5.9%; $t_{(16)} = 2.2$, $p < 0.05$).

**Confidence in updated memories correlates with PE.** Going beyond the binary measure of Test2 accuracy, we explored the subjective assessment of memory accuracy, as expressed by Test2 confidence ratings. We posited that upon correction by feedback, incorrect answers expressed with high confidence in Test1 (large negative PE) would be replaced by correct answers recalled with high confidence in Test2. Conversely, questions left unanswered, or incorrectly answered with low confidence in Test1 (small negative PE), would be answered with low confidence upon correction in Test2 (Methods). We found that when participants successfully learned from the feedback, confidence in the correct Test2 answer was correlated with the PE in Test1, such that larger negative PE values in Test1 were associated with greater confidence in the correct Test2 answers. In the recall paradigm (Fig. 2b), there was a significant main effect of PE on Test2 confidence ($F_{(3,57)} = 3.7$, $p = 0.017$), and a significant difference in the high vs. low PE conditions (mean 58.9 ± 3.6% vs. 74 ± 5%; $t_{(19)} = 3$, $p < 0.05$, Bonferroni corrected). In the recognition study, individual linear-regression models also revealed a significant predictive relationship between PE and Test2 confidence (mean $\beta = 0.29 ± 0.05$; $t_{(26)} = 5.8$, $p < 0.001$). A group-level regression

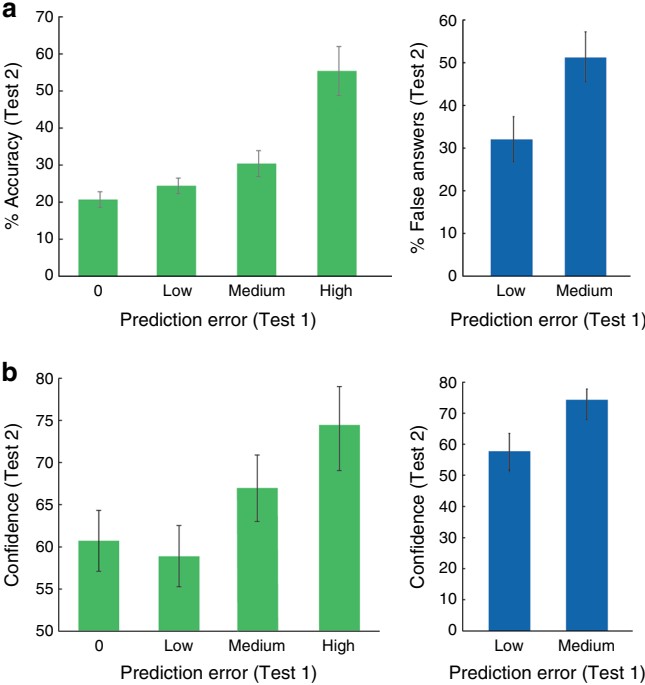

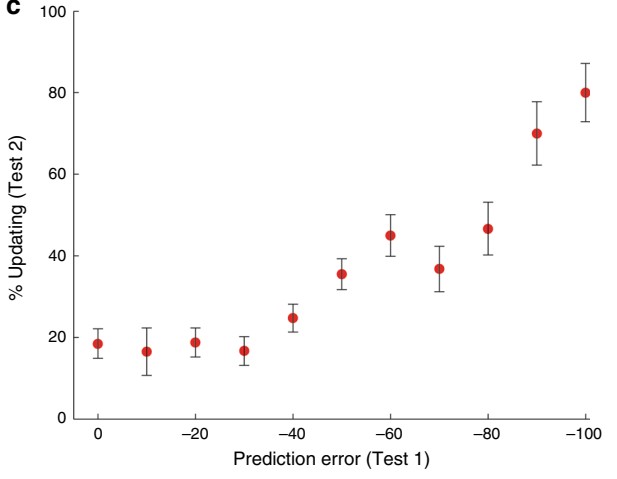

**Fig. 2** Behavioral results—PE directly determines the degree of learning from feedback and adoption of false memories. **a** Subsequent accuracy in the recall study. For incorrectly answered Test1 questions (green), the greater the PE elicited by feedback, the more likely it was for that information to be incorporated into memory, as shown by average Test2 accuracy (left). Similarly, for correct Test1 answers (blue), false-feedback information was more likely to supplant an accurate memory recalled with medium confidence (medium PE) relative to one recalled with low confidence (low PE), as shown by the average percentage of false-feedback answers subsequently supplied in Test2 (right). **b** Subsequent confidence of updated memories in the recall study. PE also determined the confidence expressed in correct Test2 recollections, for questions initially answered erroneously in Test1. Thus, when successfully learning from feedback, incorrect Test1 answers stated with high confidence (high PE feedback) were associated with high-confidence correct answers in Test2 and vice versa (average Test2 confidence for incorrect-to-correct answers on the left). Similarly, for the subset of questions answered correctly and supplanted with false feedback, average Test2 confidence in the false memory was greater when initial confidence in the correct answer was higher (medium vs. low PE, on the right). **c** Summary of behavioral results in the recognition fMRI study. A strong positive relationship was observed between the degree of overall memory updating and PE arising from feedback, for questions answered incorrectly in Test1. This measure of learning combines both subsequent accuracy and confidence (see Supplementary Figure 3 for separate measures). Error bars represent SEM

model of the median confidence scores (Supplementary Figure 3) indicated that each unit increase in magnitude of the negative PE led to a corresponding increase in Test2 confidence of nearly half a percent ($\beta = 0.43$, intercept $= 45$, $R^2 = 0.89$, and $p < 0.001$). Thus, median confidence for 0 PE trials was 47.5%, as opposed to 90% for the $-100$ PE trials. Analysis of correctly answered trials also revealed a significant relationship between PE and confidence updating (Supplementary Figure 4).

The same phenomenon was observed in the false-memory condition for the subset of correct-to-false trials (Fig. 2b). Participants expressed greater confidence in their newly acquired false memories, when the false-feedback information was adopted in place of accurate memories (Test1 correct answers) of medium confidence, compared to the confidence in false memories which supplanted low-confidence correct answers (mean $74 \pm 3.62$% vs. $57.8 \pm 5.27$%; $t_{(81)} = 2.5$, $p < 0.05$).

**A general PE-based learning effect.** The results demonstrate two forms of memory updating. The prediction error was not only related to the likelihood of successful learning and subsequent recall of the feedback information (memory content/accuracy), but also to the subjective strength of this learning (metamemory confidence). We devised a method by which the two metrics could be combined to enable a holistic measure of memory updating (Methods). In the recognition study (Fig. 2c), the group-level regression shows that each unit increase in PE led to a 0.6% increase in this measure of updating ($\beta = 0.61$, intercept $= 6.95$, $R^2 = 0.85$, and $p < 0.001$). Hence, overall updating from 0 PE feedback was 18.5% vs. 80% from $-100$ PE feedback. This effect was also significant at the subject level (mean $\beta = 0.27 \pm 0.06$; $t_{(26)} = 4.13$, $p < 0.001$).

**Parametric brain encoding of informational PEs and salience.** In theorizing about how the brain might respond to the semantic PEs in our task, we posited that there may be regions which register the magnitude of the prediction error alone, but are not concerned with the valence of the PE (absolute PE values), and regions which register both their valence and magnitude (signed PEs; Supplementary Figure 6). In the reinforcement literature,

unsigned PEs are synonymous with salience. Accordingly, we ran two analyses to establish whether this was the case (Methods).

**Regions correlating with both valence and magnitude of PE.** The first analysis revealed a highly symmetrical, bilateral network of regions whose activity correlated with a continuous increase in PE, from high negative to high positive values (Fig. 3 and Supplementary Table 1), consistent with an encoding of signed PEs. The largest and most significant clusters were observed in the striatum, cingulate cortex (CC), dorsolateral prefrontal cortex (DLPFC), inferior parietal cortex, and precuneus. Within the striatum, there were two large clusters of activity: the first, centered on the VS, incorporating nucleus accumbens (NAc), was the most powerful, and extended dorsally up the caudate nucleus; the second cluster incorporated large areas of the dorsal and ventral putamen—particularly lateral and posterior parts—and extended laterally through the claustrum to the insula. On the right hemisphere, the putamen cluster also extended ventrally and anteriorly to the anterior MTL, including amygdala and HC. Activity in the CC appeared to be related to the striatal activations, similarly occurring in two major clusters: a mid-anterior cluster and mid-posterior cluster, both centered at the same location on the $y$-axis as the VS/NAc and putamen/claustrum clusters, respectively. The large DLPFC clusters were centered on the middle frontal gyri, with some smaller clusters in the superior gyri, and the large parietal activations were observed in the inferior parietal lobule (IPL). The contrasts also revealed smaller bilateral activations in the fusiform gyrus, occipital cortex, and cerebellum.

No regions were found to be inversely correlated with PE, i.e., to be the most active for large negative and least active for large positive PE events.

**Regions correlating with absolute PE magnitude.** The second analysis revealed regions with a V-shaped response to the range of PEs encountered (large positive activations to high PEs of either valence), implying an encoding of the unsigned PE magnitude, or salience/surprise (Fig. 4; Supplementary Table 2). Two main bilateral clusters of activity were found. The first was a large activation with peak activity in the inferior frontal gyrus (IFG), also encompassing areas in the anterior insula and anterior claustrum. The second was observed in the dorsomedial prefrontal cortex (DMPFC), with peak activity located on the medial surface of the superior frontal gyrus. There were two smaller clusters located in proximity in the medial frontal gyrus, as well as two clusters in the right DLPFC.

Additionally, several regions exhibited activity consistent with an inverse V profile of response that was greatest to small PEs and smallest to large PEs, of either valence. These were mostly located in the MTL, in mid, and posterior parahippocampal gyri, bilaterally. The posterior parahippocampal deactivations also extended to parts of the lingual gyri and posterior cingulate. The more anterior deactivations included voxels in the HC on the left hemisphere. Another large cluster was observed on the left side at the intersection of the parietal, temporal, and occipital lobes.

**PE-responsive regions are predictive of subsequent memory.** We hypothesized that brain regions which register the PE of the feedback information are likely to be involved in the encoding of that information, or updating of prior knowledge. We therefore performed further analyses to link those activations with the behavioral memory effects.

We found activity significantly predictive of confidence updating within the signed PE regions (Fig. 5a and Supplementary Table 1), but not in any of the unsigned PE regions. Highly significant correlations were observed bilaterally in the VS and

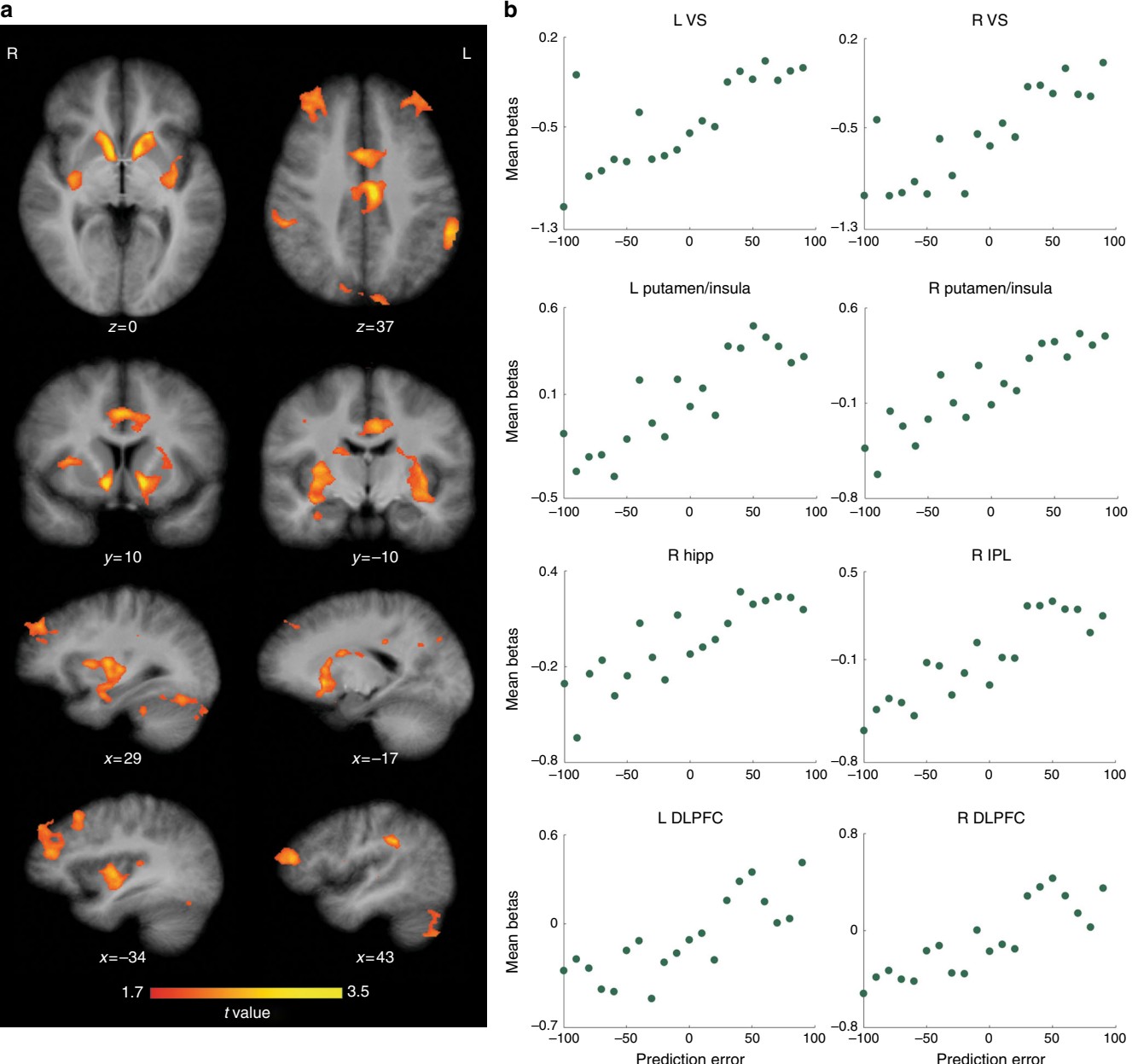

**Fig. 3** Brain regions correlating with PE magnitude and valence during feedback. **a** Substantial clusters of activity were observed bilaterally in ventral and dorsal striatum, DLPFC, cingulate cortex, inferior parietal cortex, and precuneus (Supplementary Table 1). Within the striatum, one cluster was centered on the NAc and another in posterior areas of the putamen/claustrum, extending through to insula and (on the right) anterior MTL. **b** Activity in these regions parametrically varied with the signed PE on each trial, such that the greatest responses were elicited by better-than-expected outcomes (feedback for correctly answered questions stated with low confidence—large positive PE events), and the smallest responses by worse-than-expected events (feedback for incorrect answers stated with high confidence—large negative PE). Importantly, responses to fully predicted outcomes (zero PE feedback, whether for correct or incorrect answers) lay in the middle. The plots depict average single-trial beta estimates for each PE value, within selected ROIs

DLPFC ROIs. Additionally, the DS (caudate) bilaterally, left fusiform gyrus, and several smaller PFC ROIs also exhibited significant correlative activity. These correlations were negative, such that greater deactivation during feedback for incorrect answers was associated with higher confidence in subsequently corrected Test2 answers. Conversely, activity in these (and other PE) regions was not significantly correlated with Test2 confidence for incorrect-to-incorrect trials (Fig. 5a), further demonstrating their specific significance with respect to the updating process.

These analyses also revealed four regions whose activity during feedback was predictive of subsequent accuracy (Fig. 5b and Supplementary Table 1). Three of these—the mid/ posterior cingulate, right IPL, and right putamen/insula cluster—were of the major clusters identified in the analysis of signed PE regions. The other was in the right inferior lateral PFC. In each of these regions, greater deactivation in response to negative feedback was associated with a higher likelihood of subsequently answering the question correctly in Test2. A whole-brain analysis of incorrect trials (updated > not updated) did not reveal any regions that were predictive of content updating. As with the confidence updating, a separate analysis revealed that in none of the unsigned PE ROIs did activity differentiate subsequently corrected from uncorrected answers.

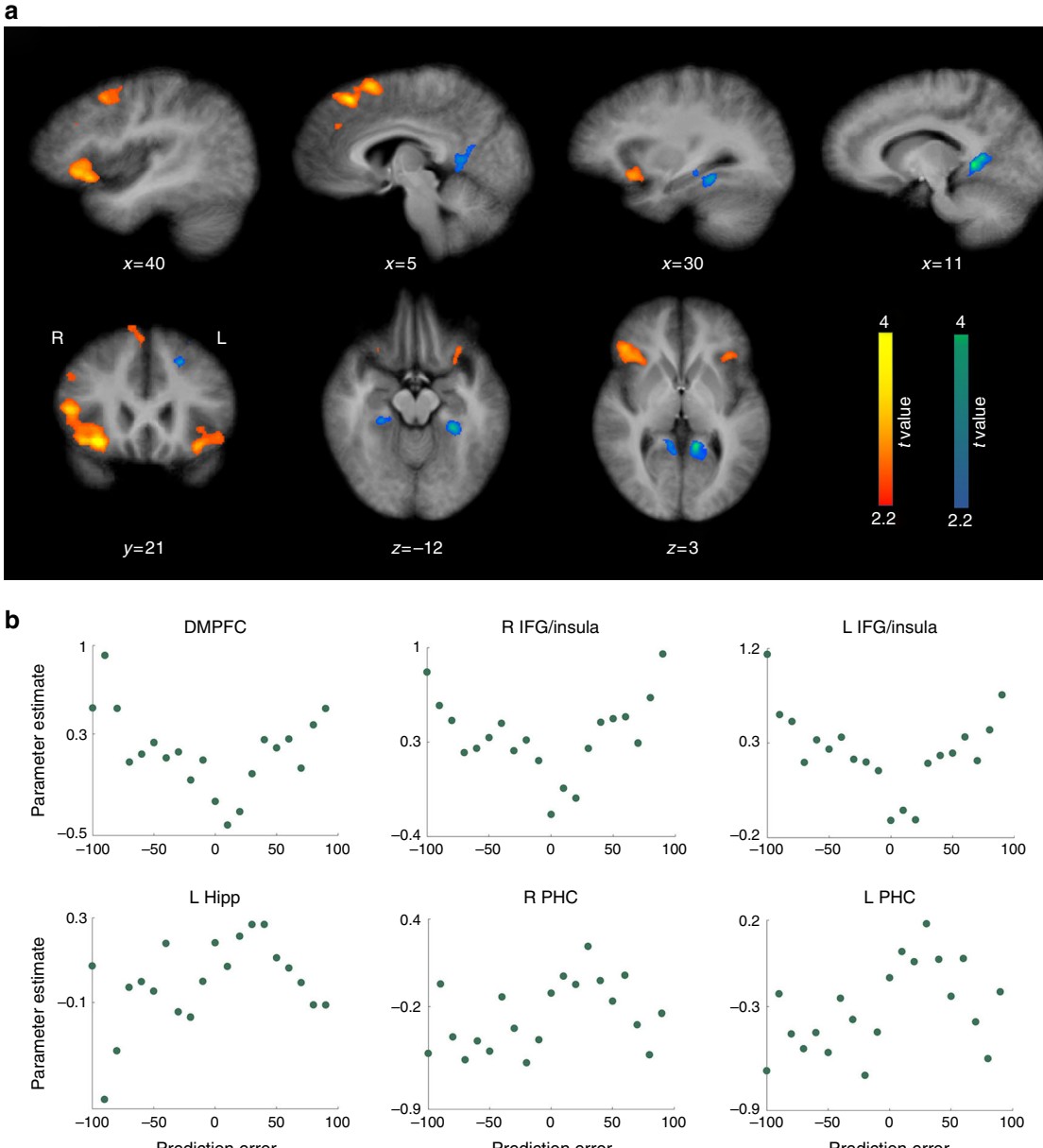

**Fig. 4** Brain regions correlating with absolute PE magnitude (salience). **a** Large clusters in IFG (extending to anterior insula) and DMPFC bilaterally, as well as DLPFC (on the right) were observed to positively correlate with unsigned PE magnitude. A number of activations (in blue) in the MTL, mainly in the parahippocampal gyri (PHC), inversely correlated with unsigned PE magnitude (Supplementary Table 2). **b** Activity in these regions parametrically varied with PE magnitude on each trial, such that the greatest responses were elicited by large PE events of both positive and negative valence (feedback to correct answers stated with low confidence, and incorrect answers with high confidence), and the smallest responses by low PE feedback events (following high confidence correct, and low confidence incorrect answers). The reverse relationship was observed in the blue activation clusters. Plots depict average single-trial beta estimates for each PE value in selected ROIs

## Discussion

The updating of long-term memories by integrating new information into existing knowledge is essential to maintaining the relevance and predictive utility of one's knowledge base in an uncertain and dynamic world. Our results provide evidence that a Rescorla–Wagner-type model of prediction-error-based updating can be applied to declarative learning, whereby new information is compared with the brain's existing beliefs and expectations to establish its informational value, thereby determining the strength of its encoding and integration with prior knowledge.

We observed behaviorally that the degree of memory updating in response to feedback information directly depends on the PE of that information, as it relates to varying degrees of expectancy violation. Growing recognition that memory systems operate in an integrative manner has inspired numerous efforts to demonstrate the influence of reinforcement and PE-type processes on episodic recognition memory for simple stimuli[20,23,25,27,30–34,42,43]. However, the source of reward, or PE, in these paradigms is typically incidental to the to-be-remembered stimuli, and differences in memory are often weak. In the current paradigm, the information itself is the source of the PE—a critical feature of the design which enabled us to show commonality in the rules of learning, as opposed to an influence of one system on another.

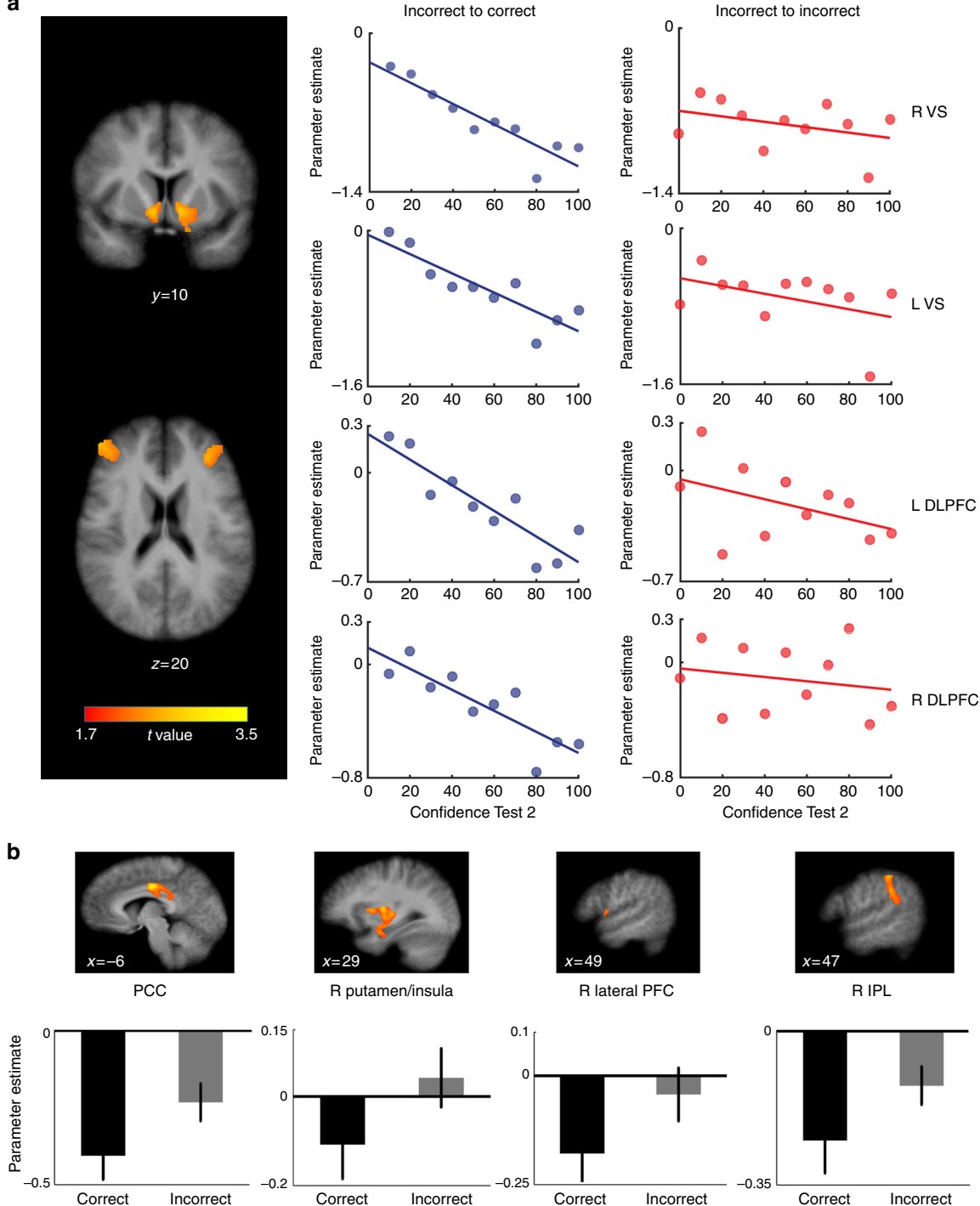

**Fig. 5** PE encoding regions predictive of memory updating. **a** For the subset of questions answered incorrectly in Test1 and accurately in Test2 (i.e., successful learning from feedback), confidence ratings in Test2 answers were highly correlated with responses observed in the VS and DLPFC ROIs during feedback (Supplementary Table 1). On the right, average single-trial beta plots for these ROIs show that the greater the reduction in activity elicited by negative feedback in Test1, the stronger was the confidence in correct (blue), but not incorrect (red) Test2 answers. **b** For all questions answered erroneously in Test1, the response to negative-feedback information in cingulate, right IPL, and right putamen PE clusters, was predictive of subsequent accuracy in Test2 (Supplementary Table 1). Greater deactivation in these regions was associated with feedback that was subsequently incorporated into memory vs. feedback that was not (incorrect-to-correct vs. incorrect-to-incorrect trials). Subsequent memory correlative activity was not detected in unsigned PE ROIs

The putative PEs varied in valence and magnitude, and were rich in the sense that they were semantic rather than perceptual in nature—predictive of long-term memory in proportion to their magnitude.

In the case of erroneously answered questions, the result that memory updating from feedback is related to its PE closely resembles the hypercorrection effect. This paradigm involves a test of general knowledge where confidence ratings are measured,

followed by an immediate retest. Several studies have shown that high-confidence errors are more likely to be corrected relative to low-confidence errors[28,44,45] and are associated with P3-like potentials in EEG[44]—similar to those elicited by "oddball" stimuli —and ACC, DLPFC, and TPJ activations in fMRI[45]. The dominant explanation posited in this literature is that the surprise arising from high-confidence errors rallies attentional resources, thereby enhancing learning. Our PE-based explanation differs in that it reveals features inherent to the rules of declarative learning itself. The VS response we observed supports this approach, both because this region is not typically associated with attention, and because the response profile linearly tracked the signed PE, whereas an attentional response would not differentiate surprising outcomes based on their valence. These two approaches are reminiscent of the Rescorla–Wagner vs. Pearce–Hall debate, in which changes in associative strength are directly driven by PEs in the former and in the latter result from error-based modulation of attention[46]—but they are also compatible and may both provide explanatory utility here. For instance, the salience response we observed is consistent with the attentional explanation (see below); notably, this network did not include striatal regions. Interestingly, only in the signed PE regions were the activations predictive of subsequent memory, which suggests that PEs may be more important in mediating the updating effect than a generic attentional response alone. Procedural and methodological differences may account for why no striatal activation was observed in the hypercorrection study. These include the design of the fMRI analyses and behavioral measures, a significant temporal separation between the test (prior to scanning) and feedback (provided in the scanner), and the study-test–test design with novel material employed here vs. the test–test general knowledge protocol.

Additional evidence for a PE approach to semantic memory was deduced from the confidence in updated memories. This effect has not been previously documented and highlights the different aspects of updating which are determined by PE: memory content and metamemory assessment. Yet, the most startling and counterintuitive of the results predicted by our hypothesis was that stronger accurate memories were more amenable to being supplanted by false (conflicting) information than weaker ones, and that confidence in newly acquired false memories was greater when they replaced accurate memories expressed with greater confidence. The literature on false memory is now replete with paradigms designed to distort memories of experienced episodes, or implant entirely fabricated ones[47]. Much of the research on false memory has focused on the conditions under which people are susceptible to the impact of misleading post-event information, such as in the misinformation effect[48]. Remarkably, we were unable to find any previous studies showing a similar effect of enhanced misinformation adoption for subjectively stronger memories, or in general pertaining to the relationship between memory confidence and susceptibility. However, we caution that this result was based on a small number of trials and warrants further study.

Pertinent here is the view that consolidated memories are not stable, as once thought, but dynamic and malleable, with the ability to be modified by new or conflicting information[49]. The foremost paradigm by which this is demonstrated is reconsolidation, whereby recollection, or reactivation of a memory can induce a temporary lability, rendering it vulnerable to alteration[50]. However, recent discoveries have shown that reactivation alone is neither necessary nor sufficient to destabilize memory. What does appear to be critical is the concomitant generation of a (dopamine-dependant) PE, by the introduction of new information, or by expectancy violation[50,51]. These findings suggest that the purpose of reconsolidation is memory updating, which is only

of value when there is a need to modify a memory in light of new information. Although there is currently only limited evidence that PE is a determining factor for reconsolidation in human declarative memory[50,52], the fact that post-event paradigms of memory change typically contain this element indicates that the parallels are relevant.

The applicability of trial-and-error-based rules of learning to knowledge acquisition does not prima facie seem fitting. However, learning always takes place upon a backdrop of prior experiences, semantic schema, and beliefs, against which incoming information can be judged—even when it is entirely novel. Semantic knowledge is also formed over cumulative exposures to information and shares with conditioning and procedural memory, a reliance on slow encoding of rigid associations[41]. This view of knowledge acquisition is parsimonious with predictive coding models of brain function[1,2,39], folding both the initial learning and subsequent updating into one scheme based on surprise reduction. In this light, retrieval-based phenomena such as the misinformation effect and reconsolidation are entirely in accordance with the general principles of learning.

One limitation of our study relates to the method of utilizing confidence judgments as a proxy for memory strength, or degree of expectation, to estimate declarative PEs. Another critical determinant of PE magnitude is likely the degree of accuracy (or rather inaccuracy) of the recalled memory relative to the information in the (feedback) answer—that is the veracity of the memory in terms of its semantic content. This discrepancy is perhaps a more direct measure of outcome–expectation in the classic sense, but is difficult to estimate mathematically and is highly subjective.

Critical to our endeavor was the observation of a neural correlate of the informational PEs entailed in our paradigm. We discovered a network of regions which appeared to powerfully respond to the feedback information, in a manner strikingly consistent with a PE signal—encoding the discrepancy between expectations based on existing knowledge and new information. Of primary interest were the large and powerful PE responses observed in the striatum, particularly the VS, which is most commonly associated with reward PEs in fMRI studies of reinforcement learning[5,8–11], and a major projection site of the dopamine neurons thought to encode them. To our knowledge, a direct correlate of PEs has not been previously demonstrated in declarative learning of this nature, and few cognitive PE correlates, such as in perceptual associative learning, have ever been observed in the striatum[3,20,53].

There is now substantial evidence for a role of the striatum in declarative memory encoding. This has primarily been shown in the context of recognition memory for words or pictures presented during or after reward-associated stimuli, or motivated by future reward[31,32,42,43]. For example, picture cues predictive of monetary reward are subsequently recognized better than neutral cues, and are associated with greater activity in the caudate and midbrain[33]. The caudate is also engaged by feedback during declarative associative learning[54]. Subsequent memory-related striatal activations have also been observed in experiments devoid of reward or PE. For instance, increased putamen activity and MTL connectivity were shown during the encoding of subsequently recognized vs. forgotten words[30], and the caudate exhibits a memory-predictive signal at the offset of short film clips[29].

In addition to registering the declarative PE, we also observed a role for the same striatal regions in subsequent memory, such that their activity differentiated successfully from unsuccessfully encoded feedback, and confidence in the former. However, unlike previous studies[30,34,54], we observed greater memory updating as a function of decreasing striatal responses during our behaviorally relevant condition of negative feedback. Although some brain

regions consistently show deactivation-related subsequent memory effects (see below), this is a perplexing result if dopamine-enhanced long-term potentiation is the mechanism by which PE modulates memory[26]. On the other hand, the relationship between negative/punishment PEs, dopamine firing, and striatal activity is a complex one. A hotly debated question is whether these regions respond in a valence-specific manner to PEs or instead signal "salience"—a loosely defined term relating to the significance of a stimulus regardless of its valence[5–7,55]. Dopamine neurons and striatal regions have also been shown to increase their firing rate/activity to unexpected stimuli of both valences, and to affectively neutral salient stimuli[7,8,23,43,55,56]. Since we did not observe midbrain PE correlates in this study, we cannot determine the consequences of these striatal deactivations with respect to the dopamine-mediated mechanism of memory enhancement[26,31].

An important point to consider is whether VS activations in our task reflect discrepancies between expected and observed information, or instead are indicative of reward PE responses associated with goal attainment. If purely cognitive feedback relating to task performance is intrinsically motivating, reward and punishment responses may be expected in tasks that, like ours, do not involve extrinsic rewards. Probabilistic classification learning and visual categorization studies are illuminating in this regard, since no extrinsic reinforcement is utilized, and feedback is only indicative of response accuracy. Such studies have yielded varying, inconclusive findings as to the response profile of VS to performance feedback[57–60]. Often, both positive- and negative-feedback stimuli increased VS activity. Feedback-based caudate activity has also been observed in declarative learning with a paired associates word task, but in the DS, where again, the response profile (from negative to positive) is unclear, varies over studies, and depends on factors such as the number of choice options and stage of learning[54,61,62]. Thus, it would not appear that cognitive, performance-related feedback in itself (i.e., devoid of semantic content) is akin to value-based reward and punishment, nor accounts for the bivalent VS responses we found, which resemble typical reinforcement-evoked responses. In general, striatal responses to performance feedback are observed where task performance is emphasized, either explicitly, or where there is an expectation of improvement over trials. Additionally, the feedback typically comprises very simple valanced cues. Conversely, in our study, the participants were told that they would likely answer many questions incorrectly, nor was there any ability to improve over trials, and the feedback was not overtly positive or negative but required semantic processing to evaluate the veracity of prior answers.

Our findings therefore suggest that new information, as it relates to the unexpected confirmation or refutation of extant knowledge and beliefs, may be inherently rewarding or aversive, such that reinforcement is embedded in natural knowledge acquisition and updating via semantic PEs. The idea that information alone can be of value and recruit reward systems has been suggested in prior studies[43,55]. However, it remains to be seen whether the VS response would obtain outside the context of a task—itself encoding a purely semantic PE, or instead combining semantic PE signals encoded elsewhere with explicit performance-based reinforcement signals.

Beyond the striatum, we detected signed PE correlates in a bilateral network of cortical regions. It is noteworthy that activity in all of these regions has been shown to correlate with reinforcement-related PEs[10], strengthening our hypothesis that PE-based declarative and nondeclarative learning share common neural substrates. Similarly, there appears a large degree of overlap between this network and regions commonly identified with subsequent memory and forgetting during episodic

encoding[63,64]. We also noted PE correlative activity in the right anterior MTL, which incorporated the amygdala and HC. While the amygdala is implicated in encoding reward, and punishment-related PEs[10], the HC is not typically. MTL responses to the occurrence of unexpected stimuli are well documented; however, a direct hippocampal PE correlate has thus far only been demonstrated in nondeclarative probabilistic categorization learning[40], and in an episodic action–observation paradigm[20], but not in the type of declarative semantic learning studied here.

Additionally, we found absolute PE, or saliency-associated activations bilaterally in the DMPFC and anterior insula/IFG. These regions have been shown to signal saliency in several reward-based imaging studies[9,11,56], respond to salient/surprising affectively neutral stimuli[24,43], and are implicated in memory encoding[63,64]. Of most interest in the saliency network were the numerous MTL clusters, primarily in the parahippocampal gyrus which is widely observed to be predictive of successful encoding[63,64] and exhibits enhanced responses to salient stimuli[24,43]. Reward and PE-related activity has been observed in PHC numerous times in fMRI[10], and on three occasions has been explicitly demonstrative of a non-valenced PE signal[56,65,66]. In two studies, such activity was inversely correlated with outcome salience, as we found[66,67]. Moreover, parahippocampal repetition suppression effects to visual stimuli, and their subsequent recognition, are modulated by dopamine[68].

A primary function of saliency responses entails the marshaling of attentional resources to significant events. This is consistent with the increased activity in frontal regions and decreased activity in MTL, PCC, and TPJ we observed[69]. The latter are prominent nodes within the default-mode network (DMN)[70], known to deactivate with external task engagement[64,71], reflecting a shifting of attention from internal processes to external stimuli. Accordingly, the inverse V response we observed in these areas can be explained by greater attention/task-related engagement as (positive or negative) PE increased, possibly supporting the updating process.

The commonality in declarative memory encoding and PE/salience networks was further supported by a trial-by-trial subsequent memory analysis for feedback to erroneous answers. We found three cortical ROIs where feedback-related activity was predictive of Test2 veracity: right inferior parietal, right inferior frontal, and posterior cingulate—in addition to the putamen ROI discussed above. Notably, in these regions, greater deactivations in response to feedback for incorrect answers were associated with greater Test2 accuracy. This finding is particularly interesting since PCC and right inferior parietal (TPJ) regions are notable for their deactivation as being associated with superior subsequent memory for items[64,72] (and are nodes in the DMN). These two regions are also significantly connected, anatomically and functionally, to the parahippocampal gyrus—where we also observed deactivations from baseline in response to increasing PE—leading to the suggestion that the PHG serves as the nexus through which the DMN interacts with the MTL memory system[73]. Indeed, the MTL—including HC—is thought to be part of the DMN, and responds in a similar manner (deactivation in task vs. rest) in many memory studies[69,71,74]. Subsequent memory performance has even been shown to correlate with greater deactivations in MTL at stimulus presentation[75]. The inferior parietal involvement in memory is a robust finding, this region is also sensitive to perceptual violations of expectancy, correlates with reward PEs, and has been linked to violations of memory expectations[76,77]. We also observed a dissociation between regions correlating with subsequent confidence and accuracy, consistent with previous behavioral and fMRI findings[78].

These findings are prima facie at odds with many studies of declarative memory, where successful encoding is associated with

increased activity in frontal and MTL regions[64]. In a typical study, encoding related activity corresponding with the presentation of simplistic, novel episodic stimuli (pictures/words) is determined in accordance with subsequent recognition performance for those stimuli. In contrast, the memory processes taking place here involve the updating of complex semantic knowledge, which differs in three fundamental ways. First, there was previous exposure to/encoding of the information in the study phase prior to scanning; hence, a "remembered vs. forgotten" contrast for feedback in Test1 is not equivalent to looking at differential brain responses to information encountered for the first time, such as a one-shot novel picture stimulus which is subsequently recognized or not. Second, the feedback-based updating often simultaneously entailed the alteration/overwriting of a competing memory, as well as registration of a PE, rather than the pure formation of a novel memory trace. Finally, semantic and episodic memory acquisition is thought to depend on partially dissociable neural processes[12], with a greater cortical and basal ganglia involvement in the former[41]. For instance, hippocampal lesion patients with anterograde episodic amnesia can acquire new semantic knowledge[79,80]. For these reasons, attempting to view our results through the lens of the classic memory-encoding literature is best avoided.

Elucidating the determinants of efficacious, long-term memory formation, and its neural basis, is of paramount importance. This PE account of declarative memory may be of significant value in the perennial quest by educators and students to enhance retention of knowledge, by devising more effective techniques of learning and teaching.

## Methods

**Participants.** Fifty-four volunteers participated in the study, 21 in the behavioral experiment (14 males, mean age of $27 \pm 3.4$ years), and 30 in the fMRI study (right-handed, 19 males, mean age of $26 \pm 2.6$ years). One participant in the behavioral experiment was excluded from analysis due to suspicion of the false-memory manipulation. In the fMRI experiment, one participant was scanned with a different sequence at a lower resolution and was excluded from the imaging (but not behavioral) analysis. Another participant requested to exit the scanner during the second run, and a third participant was interrupted by a technical fault during the second run, leaving behavioral and imaging data for the first run alone. The experimental protocol was approved by the Institutional Review Board of the Sourasky Medical Center, Tel Aviv. All participants were healthy, had normal or corrected-to-normal vision, provided written informed consent, and were remunerated for their participation.

**Memoranda.** The documentary text was typed in Hebrew and comprised six single-sided A4 pages. The text dealt with the history of the Falkland Islands and was mostly concerned with the war that took place between the United Kingdom and Argentina in 1982. The information spanned a broad range of topics—political, military, geographical, numerical, economic, et cetera—and took most participants 35–45 min to read. This included global and local events leading up to, during, and in the aftermath of the war. We deliberately chose a topic that was arcane to our subject pool, and ensured that the information in the text was highly detailed and novel. Post-test debriefing confirmed that most participants were entirely unfamiliar with the topic. The recognition test was also piloted on a number of individuals who had not read the text, to verify that without study, a score significantly greater than chance would be unattainable.

**Experimental procedure.** On the first day (the study phase), participants were invited to a quiet room where they were asked to read the text once, at their own pace, and to try and absorb as much of the information in the text as possible for a test of the topic, 2 days later.

Participants in the behavioral study returned to the same room for Test1. The experimenter explained how to answer the questions with some practice trials (on an unrelated topic) after which they started the task. They were informed that the test would be difficult and that it was important to skip, rather than guess the answers to questions, if they had no conception of the correct answer. The cued recall test comprised 100 computer-based questions, probing their knowledge of facts explicitly detailed in the text. The same set of questions (randomized in order) was utilized for all participants.

The trial began with the presentation of the question on a screen, for example: "How many weeks did the Falklands war last?" The question remained visible while the participant typed their answer into a box presented underneath the question, pressing enter to proceed. Participants were encouraged to skip questions, rather

than make outright guesses, in which case they were instructed not to type a response and simply press enter. They were next asked to provide a confidence rating which could be any value between 0 (if skipped, i.e., no idea what the correct answer is) and 100 (indicating complete certainty that their answer was correct). The question and confidence phases were not time limited. Finally, the feedback was presented for 5.5 s, after which the next trial commenced.

During Test1, all trials had the potential to display the correct or false feedback —the latter being a plausible yet significantly different answer to the question. Just as there was only one possible correct feedback answer which could be displayed, each question was associated with one possible false-feedback answer. Naturally, this investigation necessitated that participants be naïve to the manipulation. Thus, we did not provide false feedback to any questions answered with a confidence above 70—a level determined by pilot testing to satisfy this prerequisite. Participants who nevertheless became aware of the manipulation were excluded from the study. Using a pseudo-random number generator, questions to which answers were supplied with a confidence rating below 60 displayed the false feedback on 75% of trials, and on 60% of trials with a confidence rating between 60 and 70. The large proportion of false-feedback trials was necessary because the veracity of recall answers could only be determined post testing, so we could not provide false feedback solely following correct answers. Furthermore, there were fewer correct than incorrect answers (average 36%) and they tended to be expressed with higher confidence (average confidence for correct answers was 70.5, confidence was above 70—the cutoff—in 56% of correct trials). In order to generate sufficient trials where participants were supplied with false feedback following correct answers of low and medium confidence, we therefore resorted to high proportions within these confidence ranges—the corollary being that a large proportion of incorrect answers was met with false feedback (Supplementary Table 3).

FMRI participants performed Test1 in the scanner, in two runs (50 questions in each, lasting for 20–25 min). Listed below each question were four potential answers to choose from. The same 100 questions and potential answers were used for all participants (randomized in order). Participants selected their answer using a right-hand response box featuring four buttons, or skipped the question using a response box supplied to their left hand—as with the recall study, they were instructed not to make an outright guess if they had no conception of the answer. Where an answer was selected, the chosen answer was highlighted in yellow for 0.5 s. If no response was registered within 50 s the trial proceeded. For the confidence rating, two of the right-hand buttons were used to move the red bar on the visual analog scale left or right (from an initial location of 50) in steps of 10. The confidence number corresponding to the bar's location was indicated directly above it (Fig. 1). To enter the rating and move to the next phase, the left-hand button was pressed. In the event of a mistake (incorrect button press), instructions were to enter zero for the confidence. The feedback phase displayed the question and potential answers, with the correct answer highlighted in green, and lasted for 4.5 s. There were no false-feedback answers supplied in the fMRI study. Intertrial intervals (black screen), as well as those separating question/response–confidence, and confidence–feedback phases were jittered, to aid in deconvolution of event-related neural responses (range and median: 3–7 s, 4 s; 1–3 s, 2 s; 3–10 s, 5 s, respectively).

Test2 occurred 7 days following Test1, and was a "surprise" test in the sense that participants were not explicitly told that there would be another test. The test took place in the behavioral testing room and comprised the same 100 questions (again randomized). For the fMRI participants, the four answer options were also identical to those provided in Test1 (randomized in their relative order), and selection was by way of keyboard buttons. Besides the omission of the feedback phase, all other aspects of the task were identical to those of Test1. Following the test, participants were asked to fill in a questionnaire (debrief), concerning the study, test, and their performance. We asked whether the questions were clear and they were able to remain concentrated throughout, how difficult it was to remember the details, strategies used to learn and remember details (if any), difficulty in grading confidence, noting of anything suspicious or unusual in the tests, and what they though the aim of the study was etc.

**Behavioral analysis.** Other than scoring of the typed answers to recall questions (the recall experiment), analysis of behavioral data was performed with MATLAB (version 7.14 R2012a; MathWorks) and all statistical tests were two-tailed.

We first calculated the PE for each trial in Test1. Based on the reinforcement learning literature—where the magnitude and valence of a PE is equated with the degree to which an outcome is better or worse than expected—we posited that the semantic PE valence is determined by whether the feedback information confirms or negates what is expected, or predicted by extant semantic knowledge, and its magnitude by the degree of strength (confidence) imparted in that prior knowledge/prediction. Accordingly, feedback to correct answers in Test1 stated with a confidence below 100 would elicit positive PEs (better than the expected outcome) and vice versa for incorrect answers (where confidence above 0 would imply a worse-than-expected outcome). Thus, for incorrect answers, $PE = -$confidence, and for correct answers, $PE = 100-$confidence. For example, an incorrect answer expressed with a high degree of confidence, e.g., 90% certainty that the answer was correct, would give rise to a large negative PE of $-90$, whereas an incorrect answer stated with only a 30% degree of certainty would evoke a PE of $-30$, upon encountering the feedback. Likewise, a correct answer stated with

a low degree of confidence would be associated with a larger positive PE relative to one supplied with greater certainty.

Incorrectly answered questions from the recall experiment were grouped into four bins of prediction error for each participant, and a subsequent accuracy score was calculated for each bin: 0 PE (questions which were skipped—i.e., no answer supplied, or rated with zero confidence), low, medium, and high negative PE, corresponding to erroneous answers stated with confidence values from 1 to 33, 34 to 66, and 67 to 100, respectively. To calculate a subsequent accuracy score (percentage), we summed the number of questions in each bin which were answered correctly in Test2 (i.e., incorrect-to-correct), and divided by the total number (incorrect) in each bin, for each subject. In cases where incorrect answers/skipped questions were met with false feedback, accuracy in Test2 was scored with respect to the feedback, not the original text, such that an accurate updating of memory entailed providing the feedback answer (see below).

For the subsequent accuracy analysis in the recall study, we performed a one-way repeated measures ANOVA where the dependent variable was Test2 accuracy and the independent variable was PE (0, low, medium, and high). Two missing values were substituted by the condition group mean (deletion of these participants entirely resulted in no change to the outcome of the ANOVA or post hoc comparisons). Post hoc $t$-test $p$ values were multiplied by the number of multiple comparisons (six) for the Bonferroni correction. In the recognition study, the method used to gauge confidence was by means of a visual analog scale which restricted input to units of 10 (i.e., 11 potential responses from 0 to 100). There were significantly fewer 0 PE (skipped) trials in the recognition relative to the recall test (Test1 mean of 12% vs. 29%, respectively). This facilitated a data set with a smoother distribution of PEs, enabling a more fine-grained analysis (Supplementary Figure 2). The accuracy score was the percentage of incorrect Test1 questions correctly answered in Test2, for each of the 11 PE values. The significance of the relationship between Test2 accuracy and PE was determined by performing a one-sample $t$-test on the $\beta$ values of the single-subject regressions, to show that the parameter significantly differed from zero, as well as testing the significance of the group-level regression model. To determine the effect of PE on the uptake of false feedback in correctly answered Test1 questions, a paired $t$-test was performed, comparing the proportion of false-feedback Test2 answers for low vs. medium PE bins, excluding three subjects who did not have data in one of the two conditions (mean replacement being unsuitable for a within-subjects test comparing only two conditions). The reduction in the number of bins reflected the smaller number of correct–false-feedback trials (10/36 correct trials, on average) and the narrower PE range (limited at −70). For this reason, the trials were split into two bins of roughly equal number to enable a random-effects statistical analysis, which was achieved by creating PE bins of −1 to −49 (low) and −49 to −70 (medium).

Note that the subsequent memory analysis in the recall study incorporated trials where participants skipped or erroneously answered the questions, and were supplied with false feedback (Supplementary Table 3). In these cases, accuracy of the Test2 answer was determined in accordance with the false feedback provided. When analyzing the learning of feedback following incorrect answers, we did not differentiate the two trial types (i.e., veridical and false feedback). Nevertheless, we performed an additional analysis excluding incorrect–false-feedback trials. Indeed, the PE–subsequent memory relationship remained significant, and the effect was even greater without these trials. We also ruled out the possibility that the results were influenced by unequal numbers of trials in each bin by ranking the PEs and dividing them into three groups of equal number (as well as a group of 0 PE trials). Here too, we observed a significant stepwise increase in accuracy as PE increased.

The subsequent confidence analysis in the recall study examined the subset of Test1 trials which were answered erroneously or skipped, and subsequently corrected (incorrect-to-correct trials; Supplementary Figure 1). We calculated an average of the Test2 confidence values for each Test1 PE bin and performed a one-way repeated measures ANOVA where the dependent variable was Test2 confidence (for incorrect-to-correct trials) and the independent variable was PE (0, low, medium, and high). Six missing values in the table were substituted by the condition mean. Post hoc $t$-test $p$ values were multiplied by the number of multiple comparisons (six) for the Bonferroni correction. In the recognition study, significance of the relationship between Test2 confidence and PE was determined by performing a one-sample $t$-test on the $\beta$ values of the single-subject regressions, to show that the parameter significantly differed from zero, as well as testing the significance of the group-level regression model. To test for a difference in Test2 confidence of false-feedback answers, between the subsets of low and medium PE Test1 questions answered correctly and supplied with false feedback (i.e., low PE, correct-to-false and medium PE, correct-to-false), we performed a fixed-effects $t$-test to compare pooled Test2 confidence values for each subset. This approach was taken to minimize the effect of missing single-subject data, due to the small subset of trials in this analysis.

We calculated the overall updating effect (Fig. 2c) by summing the confidence level of each correct answer in Test2 (assigning 0 to those remaining incorrect) and dividing by the total number of incorrect answers for each unique Test1 PE value. In other words, an overall score of 100 would indicate that all incorrect Test1 trials for a given PE value were answered correctly and with a confidence of 100 in Test2. If only half the trials were answered correctly, and with an average confidence of 50, the overall updating value would be 25. In the recognition study, significance of the relationship between overall updating and PE (Fig. 2c), was determined by performing a one-sample $t$-test on the $\beta$ values of the single-subject regressions to

show that the parameter significantly differed from zero, as well as testing the significance of the group-level regression model.

In the recall study, certain trial types—which occurred very infrequently—were excluded from analyses. These were instances of incorrect Test1 answers in the false-feedback condition where (a) the incorrect answer matched the ensuing false feedback (leading to a false sense of having answered correctly), or (b) the answer provided in Test2 was correct with respect to the original text (i.e., provided the correct answer despite initially failing to do so and receiving false feedback). We also excluded rare instances where correct Test1 answers were stated with a confidence rating of zero (for the false-feedback analyses), and in the imaging analyses - hence there were no +100 PE events. In the recognition study, we excluded a small number of trials where for a given question in either Test1 or Test2, (a) no answer was selected within the time limit, or (b) a mistake was made in answer selection, as indicated by a zero-confidence rating where an answer was chosen (i.e., not skipped). In instances where these events occurred in Test2, we did not exclude the corresponding Test1 trial from the imaging analysis of PE correlative regions.

**fMRI acquisition and analysis.** Scanning was performed with a 3T Trio Magnetom Siemens MRI scanner located at the Ascher Imaging Center, Weizmann Institute of Science. High-resolution T2*-weighted functional images were acquired using a multi-band-accelerated echo-planar imaging (EPI) sequence developed at the University of Minnesota CMRR (https://www.cmrr.umn.edu/multiband/), and a 32-channel head coil. This enabled the acquisition of BOLD responses from the whole cerebrum in spatial resolution of $2 \times 2 \times 2$-mm voxels (TR: 2000 ms, TE: 33 ms, flip angle: 75°, 66 slices (no gap) at an oblique angle of 30° from ACPC, and multiband factor: 3—interleaved). Following a brief practice, functional scans were acquired in two runs, after which T1-weighted high-resolution ($1 \times 1 \times 1$ mm) anatomical images were acquired for each subject with a magnetization-prepared rapid-acquisition gradient-echo (MP-RAGE) pulse sequence (TE 2.98 ms, TR 2300 ms, TI 900 ms, and alpha 9°) to allow accurate 3D reconstruction and volume-based statistical analysis.

All data were preprocessed and analyzed using BrainVoyager QX 2.8 (Brain Innovation) in combination with in-house code (MATLAB 7.14 R2012a; MathWorks) and NeuroElf (version 1.0; http://neuroelf.net/). The first five volumes from the beginning of each scan were discarded. Preprocessing of the remaining images included realignment, removal of head-motion artifacts, slice-scan timing correction, and high-pass frequency filtering. The functional and anatomical data were aligned and spatially normalized to Talairach space. The normalized functional data were spatially smoothed using a 3D 6-mm full-width-at-half-maximum (FWHM) Gaussian kernel.

The focus of the fMRI study was to identify the brain regions whose blood-oxygen-level-dependent (BOLD) activity and timing corresponded to the putative declarative PEs elicited in Test1, and determine whether the nature of these responses accorded with our assumptions about the PE magnitude and valence. To this effect, our analyses primarily focused on the 4.5-s feedback epoch, wherein the PE was presumed to be elicited.

Two random-effects general linear model (GLM) analyses were performed on the functional data, examining variance in regional BOLD response attributable to different regressors of interest, relating to PE encoding and subsequent memory. All trial events were modeled by convolving them with the canonical hemodynamic response function (HRF) for their duration. Feedback events of different trial types were modeled as separate predictors in accordance with the purpose of the GLM, and parametric modulators of various conditions were incorporated into the model as further regressors. In both GLMs, we separately modeled feedback for mistake trials as nuisance variables (such as questions where an unintended answer was accidentally selected—as indicated by a zero-confidence rating—or where no answer was selected within the time limit). During preprocessing, we identified volumes where large movements were deemed to affect the signal (causing spikes), and rows of the design matrix corresponding to these (and their adjacent) timepoints were set to zero.

The first GLM (PE encoding), modeled all events during each trial in Test1: question, response, confidence, feedback-correct (i.e., feedback following correct answers), and feedback-incorrect. The feedback regressors also incorporated parametric modulators which comprised the numerical values of the PEs as calculated above—positive for correct, and negative for incorrect feedback events. While correct Test1 responses were of limited value in terms of predicting subsequent memory behavior over the timescales entailed in our paradigm (since nearly all questions answered correctly in Test1 were also answered correctly in Test2), they were of importance with respect to delineating their underlying neural correlates. For detecting BOLD responses correlated with PE at the time of feedback, two statistical maps were produced (signed, unsigned PE), with a random-effects group analysis of the beta images from the single-subject contrast maps, identifying regions showing significant modulation by conjunctions of regressors (each significant), specified at the first (single-subject) level (see below). The contrast maps display voxels with a statistical significance of $p < 0.001$ (uncorrected) calculated according to the formula $p = \alpha^n$ where $n$ is the number of conjunctions, with a minimum cluster size of 10 voxels.

For detecting BOLD activity that correlated with a continuous increase in PE, from high negative to high positive values (reflecting an encoding of magnitude and valence—or signed PE), we performed a conjunction of three contrasts: voxels

where activity for positive PEs was greater than negative PEs (feedback-correct > feedback-incorrect), voxels where activity positively correlated with positive PEs (+parametric-feedback-correct), and voxels where activity positively correlated with negative PEs (+parametric-feedback-incorrect). Regions positively correlating with negative PEs show a profile of reduced activity as the negative PE gets larger, in other words, greater activity for less-negative values (small negative PE) and lower activity for more-negative values (large negative PE). This conjunction enabled the identification of voxels in the brain whose activity linearly increased from least active (most deactivated) in response to large negative PE events, to most active in response to large positive PEs, i.e., tracking PE from −100 to +100 (Supplementary Figure 6). This analysis was stricter than a simple model with one parametric predictor containing all PEs, which could have identified regions where, e.g., activity was only driven by a difference between correct and incorrect feedback.

From the same GLM, we also performed a conjunction of two contrasts where activity positively correlated with positive PEs (+parametric-feedback-correct) and negatively correlated with negative PEs (−parametric-feedback-incorrect). This conjunction identified voxels whose activity was consistent with an encoding of unsigned PE (absolute magnitude).

For the subsequent memory GLM, all events were modeled as in the PE GLM, except for the feedback events which were modeled by three predictors: correct, incorrect-to-correct, and incorrect-to-incorrect. Thus, Test1 feedback events for incorrectly answered questions were classified in accordance with the subsequent Test2 accuracy for those questions (rather than the PE). In addition, the incorrect-to-correct and incorrect-to- incorrect predictors incorporated a parametric modulator, which was the Test2 confidence score (0–100) supplied with each answer. From this GLM, we performed two analyses.

To determine if any PE regions were associated with the confidence-updating effect (Fig. 2a), we looked for brain activity that correlated with Test2 confidence scores during the feedback events in Test1. Of interest here were the subset of incorrect-to-correct trials. We restricted this contrast to the PE regions of interest (ROIs) delineated by the initial GLM, performing one analysis for the signed PE regions and another for the unsigned regions. To ascertain which of the PE regions exhibited a difference in activity that was predictive of Test2 accuracy, for questions answered erroneously, we contrasted the incorrect-to-correct and incorrect-to-incorrect neural responses (updated > not updated) within the PE ROIs. Mean beta values for each feedback regressor in the subsequent memory GLM were extracted from each of the ROIs, for each subject. We performed a paired t-test of incorrect-to-correct vs. incorrect-to-incorrect beta values for the subsequent accuracy contrast, and a one-sample t-test of the Test2 confidence parametric regressor beta values for the subsequent confidence contrast, each with a statistical significance of $p < 0.05$.

**Data availability**. The data that support the findings of this study are available from the corresponding authors upon reasonable request.

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

## Acknowledgements

We thank Genela Morris for comments on the manuscript. A.P. and Y.D. were supported by a Weizmann UK "making connections" grant. Y.D. was supported by the I-CORE Program of the Planning and Budgeting Committee and The Israel Science Foundation (grant no. 51/11). A.M. was supported by an Israel Science Foundation grant (no. 603/15).

## Author contributions

A.P, Y.D., and A.M. planned the study. A.P. and N.S. designed and conducted the experiments. A.P., N.S., A.B.-Y., and A.M. performed the analyses. A.P., A.B.-Y., A.M., and Y.D. wrote and revised the paper.

## Additional information

**Competing interests:** The authors declare no competing interests.

