## [Peer Review File · Nature Communications]

Reviewers' comments:

Reviewer #1 (Remarks to the Author):

In this paper the authors report on learning in response to errors in declarative or narrative information. They expose people to information about the Falkland War. Then they test them, eliciting answers and confidence levels and giving feedback (sometimes false). Then they retested. They found that performance on retest on questions that were initially answered incorrectly was correlated with confidence. This was true whether feedback was the truth or fabricated. From this they suggest learning is governed by a "declarative PE" that is similar to the reward prediction error proposed to govern associative learning by RW and other models (and to correlate with dopamine neuron firing). They then use fMRI to test for the neural signature of such an error in other subjects run in a scanning variant, finding that BOLD response in a variety of areas correlated with the putative error signal (signed or unsigned). Most remarkable amount these was probably the finding of a signed error correlate in VS, which is where the RPE is often located in scanning studies. The authors conclude that learning of declarative memory is governed by error mechanisms similar to those thought to rule reward based associative learning, and that there are neural traces of this process.

Overall I thought the study was excellent. It uses a highly creative but logical behavioral design to approach a novel, interesting, and important question – namely what are the rules and brain systems governing non-reward based associative learning (declarative memory is to me a form of associative learning). The results are generally strong and compelling, and they clearly provide important new information to a neuroscience community obsessed with RL and reward prediction errors and value learning.

In reading it, I found only one potentially significant problem in the underlying the otherwise clear and compelling idea or logic behind the study, and this was the authors contention that behavior in response to positive prediction errors was either irrelevant or could not be studied. This is actually not really expressed until the imaging data, where they say that they could not examine effects of errors on these trials because people always chose correctly on both tests. But it seems to me that even if this is the case, there is another way to look at it, which is to examine the confidence. Even if the choice is correct both times, according to the authors logic, the confidence should provide another proxy of the underlying learning, since it should increase with large positive errors. And presumably this happens unless all correct choices are 100% confident in test 1, which cannot be.

And this actually becomes quite important because at present the evidence for error based learning is from negative prediction error trials. On these trials, learning occurs with large negative errors, yet the large negative errors are confounded with low confidence. Perhaps the things that were hard to learn about in the reading phase were just hard to learn about for reasons unrelated to errors? Perhaps individuals had interference with prior information in their lives or something? This would give the same result – poor learning from test 1 for low confidence items – but for a reason other than what the authors' suggest. To rule this out, the authors need to show that learning for correct answered items, since if the authors

are correct, it should show the opposite relationship to confidence. Of particular importance in both cases is what happens to learning when someone guesses and gets it right. According to the authors' hypothesis, in these correct but low confidence cases, there would be a high positive PE and the subjects should learn very well. They will pick again in test 2 as the authors say, but now they will do it with high confidence, hopefully higher than some of their colleagues that guessed with low or maybe even medium confidence. On the other hand, if there is some interference at work, then these people will show low confidence on test 2 (maybe even making incorrect picks) just like the low confidence, incorrect choosers from test one did.....

Ok that is my only real practical objection. Now I also have some interpretive. The authors wish to call what they have found "declarative PE's", but I don't really understand what that is. And I think much of their behavioral and imaging evidence could be accounted for by suggesting that the act of getting something wrong or right in front of a tester is inherently rewarding or non-rewarding. As such, it would recruit "value prediction error" signaling mechanisms. Obviously there is a mismatch in the declarative information, but the error is ultimately a value error. And it could be accounted for simply by positing a connection between something designed to recognize that mismatch (ACC?, HC?) and the VTA dopamine system. If this is what they authors mean, they should clearly say it. If this is not what they mean, then I think that is fine but they should say more clearly what they are suggesting.

I also think the authors should give some consideration to blocking. In some regards, what they are trying to do behaviorally is to argue that learning associative information in a network that is amenable to declaration involves the same type of teaching signal that is used to learn about other kinds of associative information. Or to put it more simply, that learning is driven by more than simply contiguity. This was first shown for rewards by Kamin [1], as I am sure the authors are aware, and has been more recently demonstrated for stimulus-stimulus associative learning by Sharpe et al [2]. Here the authors are (or should be) essentially trying to show the same thing for declarative information. They should be clear and describe things that way it seems to me. It will clarify the value of what they are doing behaviorally, which I really think is quite enormous assuming it has not been done (and I do not know of it if it has).

Lastly I think the authors should consider saying something about the overtraining reversal effect. They can google this if they do not know it, but basically it is a very old literature that was obsessed with the study of why subjects do better at acquiring reversals if they have been overtrained on the initial discriminations. I think it is precisely what the authors are getting at for the paradoxical effect that learning can happen quicker in people who are sure they know the answer. And it provides again another very nice parallel with what they are finding for declarative knowledge and more basic associative learning mechanisms.

More minorly – I found the explanations of the contrasts very difficult to understand. The first part describes what sounds to me like a screen for an unsigned signal (positively correlated with both positive and negative PE's....?) and the second sounds very much like a screen for a signed signal (activity positively correlated with positive PEs and negatively

correlated with negative PE's). The rest of these sections seem to be constructed around the idea that these contrasts do the opposite of what I read them to mean, so I assume either they are written wrong or I am not understanding them. Either way, I think this needs to be fixed.

1. Kamin, L.J., "Attention-like" processes in classical conditioning, in Miami Symposium on the Prediction of Behavior, 1967: Aversive Stimulation, M.R. Jones, Editor. 1968, University of Miami Press: Coral Gables, Florida. p. 9-31.
2. Sharpe, M.J., C.Y. Chang, M.A. Liu, H.M. Batchelor, L.E. Mueller, J.L. Jones, Y. Niv, and G. Schoenbaum, Dopamine transients are sufficient and necessary for acquisition of model-based associations. *Nature Neuroscience*, 2017. 20: p. 735-742.

Reviewer #2 (Remarks to the Author):

The present study demonstrates how prediction error (PE) contributes to the updating of declarative memory (both with regard to stored information and confidence in the accuracy of that information/metacognition) through modulation of activity in striatal, cortical and limbic regions. Although PE has been shown to be influential in updating of declarative memory in many prior studies of using both general knowledge and episodic memory paradigms (unfortunately only two of which are referenced), neural evidence for the underlying mechanisms for this effect is indeed scant, although it is not as non-existent as the authors suggest. To date, there are a few event-related potential studies (i.e., Butterfield & Mangels, 2003) and one fMRI study (Metcalf, Butterfield, Habeck, & Stern, 2012, 10.1162/jocn_a_00228; notably not referenced here). Although the presence of Metcalfe et al. (2012) renders inaccurate the central claim that this study provides the first neural evidence of PE signal in a declarative memory task, to my knowledge, the authors are still accurate in stating that their work is the first to find PE modulation of the striatum in such a task. Metcalfe et al. (2012) found anterior cingulate and DLPFC activity (as well as other cortical regions), but not striatal activity, and of course ERP studies have poor spatial resolution.

The sensitivity of ventral striatal activity to signed PE in this declarative memory task is then taken as evidence that declarative and non-declarative systems share both brain mechanisms and computational (behavioral) rules. While both systems may be sensitive to PE, however, that does not necessitate that they are completely overlapping either and the authors should be careful to qualify these statements, as they can be read as going to the other extreme of arguing for complete overlap. The inverse relationship of signed PE with regions that typically show more activity during successful encoding into declarative memory does suggest that PE may engage the neural architecture of memory in a unique way that is not dependent on the type of memoranda or task characteristics (see additional comments on neural activity below), and distinct from how information is typically encoded into episodic memory.

They also distance themselves from previous work on the hypercorrection effect by

indicating that while their PE based explanation of learning is compatible with the attentional modulation proposed to strongly contribute to the hypercorrection effect, their findings reveal that inherent properties of PE are influencing learning directly. This sounds a bit like the Rescorla-Wagner vs. Pearce-Hall debate regarding the role of attention in learning theory (e.g., Roesch, Esber, Li, Daw, & Schoenbaum (2012) 10.1111/j.1460-9568.2011.07986.x). My recommendation is that if the authors wish to describe how their work is novel and differentiated from past theories on the PE effect in declarative memory, they need to present the past work on the hypercorrection effect in the introduction and describe a priori how their findings would differentiate a direct PE effect on learning from an effect mediated through attention. Simply giving a post hoc explanation and using italics for emphasis is not enough.

One of the most interesting aspects of the study was the finding that accurate memories of medium confidence (suggesting they have moderate memory strength) could be supplanted by false information (false feedback) more readily than weak memories. Although Fazio and colleagues (e.g., doi: 10.1177/0956797610371341) have shown the power of PE to correct high confidence false alarms, less work has been focused on the use of PE to instill misinformation (but see Fazio, Barber, Rajaram, Ornstein, & Marsh [2013]; doi: 10.1037/a0028649). Yet, this relatively novel component of the study is mentioned only in passing in the introduction.

In addition, I had some difficulty following how false feedback was implemented in the study. The methods section indicates that 75% of trials with a confidence rating below 60 and 60% of trials with a confidence rating between 60-70 received false feedback, regardless of initial response accuracy. Responses with confidence above 70 (i.e., high confidence) are not given false feedback. Although the rationale for not giving false feedback for confidence ratings above 70 is given on pg. 8, the choice to generally give false feedback on such a large percentage of trials is not clear, particularly since the majority of lower confidence responses were errors. Their choice of bins for false feedback described in the methods section also makes translation of false feedback rates to the PE bins in the data analysis difficult; the PE bins for Test1 errors are in thirds (1-33, 34-66, 67-100) and the PE bins for Test1 corrects receiving false feedback are in still other bins (0-50, 50-70). In the case of the correct-false analyses, the bins were chosen to make the number of responses in each bin equal, but this was not done for the analysis of information updating for incorrect responses (although there is a brief mention in the methods that if binned to have equal trials the effects were the same). In the end, it is difficult for the reader to get a good sense of how many (or the proportion) of the trials in each of the PE bins were false feedback vs. correct feedback.

Providing an analysis of "trial counts," or at least proportions of trials with true and false feedback in each of the PE bins (perhaps in supplemental), would be helpful for understanding the effects of these binning choices, as well as with interpreting the data. Indeed, regarding these counts, if only ~30% of correct trials were entered into the false feedback analysis (presumably because most correct answers had high confidence), that is only ~10-11 trials total on average (based on there being on average 36 trials correct at Test1; if 64% of trials were incorrectly answered). If these were subdivided equally into

"medium" and "low" bins, that means that only ~5 trials were in each of those bins, which is a very low number from which to derive analysis of memory updating.

Also, it was not always clear what "accuracy" meant when so many of the incorrect trials were also given false feedback. It is not accuracy in the objective sense, but in the sense of whether memory was updated with the false information, and thus, memory updating might be a better term. Notably, the "updating" term is used to only describe the dependent measure for the behavioral analysis of the fMRI study, but in that study false feedback was not given. In a related vein, I did not have a clear understanding from the main manuscript of how the updating metric for that study was calculated (Figure 2C) and why it was necessary to create this new metric, especially when the fMRI analyses themselves did not use this metric and instead used accuracy and confidence separately as DVs.

Regarding the neural correlates of PE and subsequent memory, the finding of deactivation of DLPFC and hippocampus with negative prediction errors, despite these items being better encoded, would appear to be counter to most previous studies of successful encoding in declarative memory, including studies using reward and/or novel stimuli. Additionally, the inverse relationship of the hippocampus and PHC to salience is similarly counterintuitive, as if regions classically involved in declarative memory encoding are being inhibited during acquisition of information following negative PE. The authors need to address these findings in the context of past literature more directly, as well as the possibility of dissociations in neural mechanisms for PE based encoding and non-PE-based encoding in declarative memory.

It is possible that their use of PE ROIs for these analyses is overlooking other regions that show the more expected positive relationships between activity and salience and/or subsequent memory. A whole brain analysis of subsequent memory effects as a function of PE would be a helpful addition. Finally, given the proposed relationship between VS and MTL in salience/PE-based encoding, a model that examined the interactions between these regions (or used VS as a seed region to look at dynamic causal modeling) would help to provide more sophisticated insight into how the striatal PE is influencing regions classically associated with successful encoding in declarative memory.

Minor issues:

Pg. 26: The authors indicate that most participants were entirely unfamiliar with the topic, but it is not clear how this determination was made. PE most likely becomes more important for learning in declarative memory when familiarity is low (see competing hypotheses for the hypercorrection effect). This may be particularly true for the apparent overwriting of accurate responses by false feedback, suggesting participants' inability to differentiate true and false information.

Pg. 29: Why did the authors use mean replacement for missing data in all analyses except for the comparison of medium and low PE false feedback conditions?

Pg. 29: Should be "medium and low PE Test1 questions"?

"To test for a difference in Test2 confidence of false feedback answers, between the subsets of high and medium PE Test1 questions answered corrected and supplied with false feedback (i.e., low PE, correct-to-false and medium PE, correct-to-false), we performed...

Many of the references seemed to have a spurious 'a' peppered through the author names.

Reviewer #3 (Remarks to the Author):

In this study, Pine et al. investigate whether the updating of semantic memories is determined by prediction errors, as has been suggested for other types of memory. This is an interesting question, and it is great seeing this information theoretic approach being applied to investigate how the brain deals with high-level, semantic information. The paper is well written and the methodology is sound. I have a few, mostly conceptual, comments and questions, which I detail below.

1. Negative prediction errors. As far as I am aware, negative prediction errors, in the reward literature, refer to the omission of an expected reward. However, in the current study, the author use this term to refer to the presentation of unexpected feedback. This does involve the omission of the expected feedback, but also the presentation of actual, be it unexpected, feedback. Therefore, I wonder if it might not be argued that this constitutes a positive prediction error. Could the authors comment on this?
2. Neural correlates of signed prediction error. The authors interpret the PE responses as semantic PEs. However, could it be that participants perceive correct feedback as rewarding (and vice versa for negative feedback), and that the neural responses reported instead reflect signed reward prediction errors?
3. Neural correlates of unsigned prediction error. Could this reflect an attentional effect, e.g. an alerting effect of surprising (salient) events? A reorienting of attention from internal to external signals, as a result of a surprising event, seems quite compatible with activation of frontal regions and deactivation of medial temporal regions.
4. Before reading the neural correlates section, I was under the impression that confidence updating and accuracy updating were two sides of the same coin; they seemed to be treated such in the behavioural section. How do the authors interpret the fact that these two phenomena seem to correlate with distinct brain regions?
5. For the discussion of declarative vs. non-declarative memory, in the introduction, the review paper by Henke (2010) that suggests that it may not be the type of memory (declarative vs. non-declarative, e.g.) that determines which brain regions are involved, but rather which type of neural computations are required for encoding that memory (Henke 2010), seems relevant.

Response to reviewers' comments:

Reviewer #1 (Remarks to the Author):

In this paper the authors report on learning in response to errors in declarative or narrative information. They expose people to information about the Falkland War. Then they test them, eliciting answers and confidence levels and giving feedback (sometimes false). Then they retested. They found that performance on retest on questions that were initially answered incorrectly was correlated with confidence. This was true whether feedback was the truth or fabricated. From this they suggest learning is governed by a “declarative PE” that is similar to the reward prediction error proposed to govern associative learning by RW and other models (and to correlate with dopamine neuron firing). They then use fMRI to test for the neural signature of such an error in other subjects run in a scanning variant, finding that BOLD response in a variety of areas correlated with the putative error signal (signed or unsigned). Most remarkable amount these was probably the finding of a signed error correlate in VS, which is where the RPE is often located in scanning studies. The authors conclude that learning of declarative memory is governed by error mechanisms similar to those thought to rule reward based associative learning, and that there are neural traces of this process.

Overall I thought the study was excellent. It uses a highly creative but logical behavioral design to approach a novel, interesting, and important question – namely what are the rules and brain systems governing non-reward based associative learning (declarative memory is to me a form of associative learning). The results are generally strong and compelling, and they clearly provide important new information to a neuroscience community obsessed with RL and reward prediction errors and value learning.

We thank the reviewer for the positive feedback, and appreciate the constructive comments, which led us to new insights and an improved manuscript.

In reading it, I found only one potentially significant problem in the underlying the otherwise clear and compelling idea or logic behind the study, and this was the authors contention that behavior in response to positive prediction errors was either irrelevant or could not be studied. This is actually not really expressed until the imaging data, where they say that they could not examine effects of errors on these trials because people always chose correctly on both tests. But it seems to me that even if this is the case, there is another way to look at it, which is to examine the confidence. Even if the choice is correct both times, according to the authors logic, the confidence should provide another proxy of the underlying learning, since it should increase with large positive errors. And presumably this happens unless all correct choices are 100% confident in test 1, which cannot be.

And this actually becomes quite important because at present the evidence for error based learning is from negative prediction error trials. On these trials, learning occurs with large negative errors, yet the large negative errors are confounded with low confidence. Perhaps the things that were hard to learn about in the reading phase were just hard to learn about for reasons unrelated to errors? Perhaps individuals had interference with prior information in their lives or something? This would give the same result – poor learning from test 1 for low confidence items – but for a reason other than what the authors’ suggest. To rule this out, the authors need to show that learning for correct answered items, since if the authors are correct, it should show the opposite relationship to confidence. Of particular importance in both cases is what happens to learning when someone guesses and gets it right. According to the authors’ hypothesis, in these correct but low confidence cases, there would be a high positive PE and the subjects should learn very well. They will pick again in test 2 as the authors say, but now they will do it with high confidence, hopefully higher than some of their colleagues that guessed with low or maybe even medium confidence. On the other hand, if there is some interference at work, then these people will show low confidence on test 2 (maybe even making incorrect picks) just like the low confidence, incorrect choosers from test one did.....

The crux of the reviewer’s concern here is whether confidence is tied to ‘difficulty of learning’. Confidence is often presumed to be a gauge of retrieval ease/memory strength, but if as the reviewer suggests, it is indicative of the difficulty of initial encoding, perhaps information that led to high confidence errors is simply easier to encode? This is an interesting point of debate and goes to the heart of the question at hand. To gain more insight on this we followed the reviewer’s suggestion and analyzed the change in confidence for *correctly* answered Test 1 questions. Indeed, we found in accordance with the reviewer’s hypothesis, that the gain in confidence for correct-to-correct answers was larger for the low confidence Test 1 (high positive

PEs) answers and decreased in proportion to the initial confidence value. In fact, at Test 1 confidence values of 70 we saw no improvement in this metric, and at higher levels even *decreases* in confidence, for the same correct answers on Test 2 (despite having had confirmatory feedback in Test 1). Bear in mind there were no guesses, since participants were explicitly instructed not to guess – hence there were no instances of zero-confidence correct Test 1 answers. We have now included these results in the manuscript results section and supplemental figures, along with the rationale for performing this analysis, written as follows:

“Finally, we also performed an analysis of confidence updating in trials answered correctly in both tests, to see whether there was a relationship between Test1 PE and change in confidence across tests. When looking at the difference (confidence Test2 – confidence Test1) we found that the greater the PE in Test1, the greater was the change in confidence. Thus for trials answered correctly with low confidence (large positive PE) a large increase in confidence was observed in Test2, whereas for high confidence correct (low positive PE) trials this metric diminished and even became negative, reflecting a decrease in confidence in Test2 (Supplementary Fig. 4; group level regression $\beta = 0.5$, intercept = -8.7, $R^2 = 0.98$, $p < 0.001$; subject level regression mean $\beta = 0.58 \pm 0.042$; $t_{(26)} = 13.6$, $p < 0.001$). This result served as an important validation of the PE hypothesis over an alternative hypothesis that the updating effects observed in the incorrect trials could be explained by low confidence being reflective of greater difficulty of learning that information.”

And in the legend to Supplementary Fig. 4:

“Supplementary Figure 4. Relationship between confidence updating and Test1 PE for trials answered correctly in both tests of the recognition study.

A significant relationship was observed between PE on Test1 for correctly answered questions and change in confidence expressed in the same answers in Test2. As PE magnitude increased (from high to low confidence in correct Test1 answers; small to large positive PE), confidence updating likewise increased (see results). Thus, questions answered correctly with low confidence in Test1 (high PE) were associated with large increases in confidence in Test2, whereas those answered with high confidence in Test1 (low PE) were associated with smaller increases, or even decreases in confidence in Test2 (despite having confirmatory feedback in Test1). Note there were no Test1 trials answered correctly with zero confidence (equivalent to +100 PE). Displayed is group average confidence in Test2 – Test1 \pm SEM.”

One final thought in relation to this question: the false memory results may be insightful here since the feedback information was novel. This would make it more difficult to argue that the effect we observe relates to the difficulty of encoding of specific information, since we observed a PE dependent updating effect based on the previous correct memory for novel (false) information, which does not rest upon the difficulty of encoding the original information.

Ok that is my only real practical objection. Now I also have some interpretive. The authors wish to call what they have found “declarative PE’s”, but I don’t really understand what that is. And I think much of their behavioral and imaging evidence could be accounted for by suggesting that the act of getting something wrong or right in front of a tester is inherently rewarding or non-rewarding. As such, it would recruit “value prediction error” signaling mechanisms. Obviously there is a mismatch in the declarative information, but the error is ultimately a value error. And it could be accounted for simply by positing a connection between something designed to recognize that mismatch (ACC?, HC?) and the VTA dopamine system. If this is what they authors mean, they should clearly say it. If this is not what they mean, then I think that is fine but they should say more clearly what they are suggesting.

This is a very important point of interpretation (also raised by Reviewer 3 below), which we spent considerable time pondering and we are indeed working on a follow-up study to tease apart the various possibilities. We examined the literature on performance related cognitive feedback - whether it is intrinsically motivating and can engage the reward PE system in the manner we observed here. Our conclusion from the fMRI literature was that task related performance is unlikely to fully account for our observations. In the first draft of our manuscript, a discussion of this issue reached nearly two pages and we felt it became too much of a diversion. However, given that two of the reviewers have debated this question, we now reintroduced this discussion into the revised manuscript in a shorter form, which we hope will make our position clearer on this question:

“An important point to consider is whether the feedback linked VS PE activations in our task reflect the discrepancy between expected and observed information, as we hypothesise, or instead are indicative of reward (PE) responses associated with (expectation-outcome mismatches of) goal attainment. It is thought that purely cognitive feedback relating to task performance can be intrinsically motivating and engage the reward system. This has been investigated in probabilistic classification learning studies where no extrinsic reinforcement is utilized and feedback following each trial only indicates whether the response was correct or incorrect. Early studies^{71,72} showed that both positive *and* negative feedback increased VS activity, indicating that cognitive feedback is not akin to value based reward and punishment. In other visual categorization studies, positive feedback has been shown to be associated with increased striatal activity relative to negative feedback. In one case both types of feedback led to above baseline activity⁷³, in another, which also included monetary reinforced trials, negative feedback was not associated with any change in the striatum⁷⁴, again showing that our findings are not consistent with performance based feedback activations devoid of semantic content. A PE (signed) correlate has however been observed in the DS and putamen (but not VS) in a non-rewarded categorization task⁷⁵, this could be accounted for by the updating of stimulus-response associations inherent to the task, rather than performance based reinforcement^{76,77}.

Feedback based caudate activity has also been observed in declarative learning in the absence of extrinsic reinforcement, but in the DS. For example, in a paired associates word learning task the caudate head was more activated by correct than incorrect feedback⁷⁸, although another caudate region (body) showed the reverse pattern, and in a subsequent study⁷⁹ positive and neutral feedback did not differ, whereas negative feedback was associated with a reduction in caudate activity and an increase in lentiform nucleus activity. Intriguingly, this caudate involvement only became apparent on the second and third rounds of learning⁷⁸, indicating that feedback itself was not reinforcing without prior expectation, suggesting a PE type response. In a subsequent study, the caudate was more activated by correct than incorrect feedback (but both positively activating), however, this difference was not apparent in a two-choice variant of the task, where both types of feedback equally activated the caudate⁶⁷. The authors suggest that this activity may have been reflective of the informational value of the feedback (which was equal in the two-choice but not in the four-choice variant), rather than reflective of task performance or reward.

In general, reward responses to cognitive feedback are observed where motivation to perform is enhanced by an emphasis on task performance, either explicitly, or due to the nature of trial and error based paradigms where there is an expectation for improvement over trials. For example, differential feedback responses to positive and negative feedback on a card guessing task were observed in the DS (and not VS), but only when correct performance was financially rewarded⁸⁰. Additionally, the feedback typically comprises very simple cues which do not require deep semantic processing. Conversely, in our study the participants were told that they would likely answer many questions

incorrectly, nor was there any ability to improve over trials, and the feedback was not overtly positive or negative but required deep semantic processing in order to evaluate the veracity of prior answers. To summarize, the literature does not support the notion that the bivalent responses we observed in the VS reflect performance based reinforcement PEs extrinsic to the feedback information. Our findings suggest that new information, as it relates to the unexpected confirmation or refutation of extant knowledge and beliefs, may be inherently rewarding or aversive, such that cognitive reinforcement is embedded in natural knowledge acquisition. However, it remains to be seen whether the VS would respond in the way we observed outside the context of a task – itself encoding a purely semantic PE, or instead combining semantic PE signals encoded elsewhere with explicit performance based reinforcement signals.”

I also think the authors should give some consideration to blocking. In some regards, what they are trying to do behaviorally is to argue that learning associative information in a network that is amenable to declaration involves the same type of teaching signal that is used to learn about other kinds of associative information. Or to put it more simply, that learning is driven by more than simply contiguity. This was first shown for rewards by Kamin [1], as I am sure the authors are aware, and has been more recently demonstrated for stimulus-stimulus associative learning by Sharpe et al [2]. Here the authors are (or should be) essentially trying to show the same thing for declarative information. They should be clear and describe things that way it seems to me. It will clarify the value of what they are doing behaviorally, which I really think is quite enormous assuming it has not been done (and I do not know of it if it has).

Lastly I think the authors should consider saying something about the overtraining reversal effect. They can google this if they do not know it, but basically it is a very old literature that was obsessed with the study of why subjects do better at acquiring reversals if they have been overtrained on the initial discriminations. I think it is precisely what the authors are getting at for the paradoxical effect that learning can happen quicker in people who are sure they know the answer. And it provides again another very nice parallel with what they are finding for declarative knowledge and more basic associative learning mechanisms.

We thank the reviewer for the blocking analogy and for the reference demonstrating the same effect in stimulus-stimulus learning. We are also grateful for the pointer to the overtraining effect which indeed provides a very nice parallel with our results. Both of these effects are now discussed in the revised manuscript along with the appropriate citations:

“...That stronger memories (erroneous or veridical) are more susceptible to being overwritten, is however reminiscent of the overtraining reversal effect – an old and paradoxical finding in the discrimination learning literature whereby subjects who receive extensive (over) training on a discrimination task are able to more rapidly learn from, and adapt to a reversal in contingencies⁵⁰.”

“...One could push the analogy further if information stored in declarative memory is thought of as an associative network which relies on a teaching signal to form and modify associations between nodes. Just as the phenomenon of blocking⁵⁷ demonstrated that the teaching signal is not solely based on contiguity in reinforcement (and more recently associative stimulus-stimulus⁵⁸) learning, the same could obtain for certain declarative associations. Thus, in process based accounts of memory systems⁴⁴, semantic memory shares with conditioning and procedural memory a reliance on slow encoding of rigid associations, subserved by basal ganglia, parahippocampal gyrus and neocortex..”

More minorly – I found the explanations of the contrasts very difficult to understand. The first part describes

what sounds to me like a screen for an unsigned signal (positively correlated with both positive and negative PE's....?) and the second sounds very much like a screen for a signed signal (activity positively correlated with positive PEs and negatively correlated with negative PE's). The rest of these sections seem to be constructed around the idea that these contrasts do the opposite of what I read them to mean, so I assume either they are written wrong or I am not understanding them. Either way, I think this needs to be fixed.

Perhaps the confusion here stems from the negative PE correlations. Regions positively correlating with negative PEs show a profile of deactivation, or reduced activity as the negative PE gets *larger*, in other words greater activity (smaller deactivation) for *less* negative values (small -ve PE) and lower activity (greater deactivation) for more negative values (large -ve PE). The first contrast (signed signal) is designed to search for voxels that correlate with a linear change in PEs – from large negative PEs on the one extreme, to large positive PEs on the other. Thus, when looking for signed PE regions, we identify regions where activity increases as positive PE values increase, and decreases as negative PE values increase in magnitude, which is achieved according to the conjunction of contrasts described in the first analysis (see also Supplementary Figure 6). Likewise, in the second analysis we are interested in regions inversely correlating with negative PE magnitude since that equates to increasing neural activity as negative PE gets larger. We clarified this point more explicitly in the revised results section accordingly:

“Regions correlating with both valence and magnitude of PE

For detecting BOLD activity that correlated with a continuous increase in PE, from high negative to high positive values, we performed a conjunction of three contrasts: voxels where activity for positive PEs was greater than negative PEs (feedback-correct > feedback-incorrect), voxels where activity positively correlated with positive PEs (+parametric-feedback-correct), and voxels where activity positively correlated with negative PEs (+parametric-feedback-incorrect). Regions positively correlating with negative PEs show a profile of reduced activity as the negative PE gets larger, in other words greater activity for less negative values (small negative PE) and lower activity for more negative values (large negative PE). This conjunction enabled the identification of voxels in the brain whose activity linearly increased from least active in response to large negative PE events, to most active in response to large positive PEs, i.e. tracking PE from -100 to +100 (Supplementary Fig. 6)...

1. Kamin, L.J., "Attention-like" processes in classical conditioning, in Miami Symposium on the Prediction of Behavior, 1967: Aversive Stimulation, M.R. Jones, Editor. 1968, University of Miami Press: Coral Gables, Florida. p. 9-31.

2. Sharpe, M.J., C.Y. Chang, M.A. Liu, H.M. Batchelor, L.E. Mueller, J.L. Jones, Y. Niv, and G. Schoenbaum, Dopamine transients are sufficient and necessary for acquisition of model-based associations. *Nature Neuroscience*, 2017. 20: p. 735-742.

Reviewer #2 (Remarks to the Author):

The present study demonstrates how prediction error (PE) contributes to the updating of declarative memory (both with regard to stored information and confidence in the accuracy of that information/metacognition) through modulation of activity in striatal, cortical and limbic regions. Although PE has been shown to be influential in updating of declarative memory in many prior studies of using both general knowledge and episodic memory paradigms (unfortunately only two of which are referenced), neural evidence for the underlying mechanisms for this effect is indeed scant, although it is not as non-existent as the authors suggest. To date, there are a few event-related potential studies (i.e., Butterfield & Mangels, 2003) and one fMRI study (Metcalf, Butterfield, Habeck, & Stern, 2012, 10.1162/jocn_a_00228; notably not referenced here). Although the presence of Metcalfe et al. (2012) renders inaccurate the central

claim that this study provides the first neural evidence of PE signal in a declarative memory task, to my knowledge, the authors are still accurate in stating that their work is the first to find PE modulation of the striatum in such a task. Metcalfe et al. (2012) found anterior cingulate and DLPFC activity (as well as other cortical regions), but not striatal activity, and of course ERP studies have poor spatial resolution.

Following the reviewer's comment, we now cite more of the hypercorrection literature including a recent review paper and the event-related studies which explored the neural underpinnings of the hypercorrection effect. We did also cite a wide range of other examples in the literature supporting a role of PE in modulating declarative memory (refs 22,25,27, 33-36, 44,45 in the original manuscript), so we do not downplay the importance of previous work or make as a central claim of this study that we provide the first neural evidence of a role for PEs in declarative memory. As we wrote in the abstract, and elaborated on in the paper, the aim was to provide more direct evidence for a commonality in brain mechanisms and computational rules governing declarative and non-declarative learning, particularly by searching for neural correlates of informational expectation-outcome discrepancies, and as the reviewer points out, the striatal result is a particularly notable neural finding in this regard (highlighted in the title), in addition to the 3 behavioural findings (accuracy, confidence and false memory PE related updating).

The discussion elaborates upon advances we have made here, and differences with respect to some of the previous literature relevant to PEs in declarative memory and neural findings therein. With specific regard to the Metcalfe et al., 2012 fMRI study, they reported medial/DLPFC, IPL and ACC activations associated with high versus low confidence errors which were illuminating in terms of metacognitive mismatch enhancement of attention, theory of mind, and memory/belief suppression, as possible components of the error correction process, but these results don't speak to declarative neural PE / Rescorla-Wagner type processes and the authors do not evaluate the behavioural and neural findings as evidence for that framework. We discuss this difference in interpretation of the hypercorrection effect, and the reviewer provides a nice parallel below with the Rescorla-Wagner vs Pearce-Hall debate. This leads us on to a number of important differences between this study and ours: whereas they implemented a subtraction analysis between high vs. low confidence answers, we set out to find a neural PE correlate by incorporating a parametric regressor that provided a numerical estimate of the PE on each trial, which could identify a linear response to varying degrees of positive and negative PEs (signed and unsigned). Other procedural differences such as the large delay between test (outside scanner) and feedback in their study, and the study-test-test in our paper vs. test-test protocol (i.e. general knowledge material with varying background familiarity versus novel material), are also meaningful in terms of the fMRI findings, perhaps explaining why no striatal involvement was found. Finally, one of our aims was to investigate the relationship of these PE related activations to memory/confidence updating, whereas Metcalfe et al. did not probe the behavioural effect (subsequent test performance) in terms of feedback related neural activity since the study was more concerned with psychological processes occurring during metacognitive mismatch.

Changes to the text are detailed in the related comment below.

The sensitivity of ventral striatal activity to signed PE in this declarative memory task is then taken as evidence that declarative and non-declarative systems share both brain mechanisms and computational (behavioral) rules. While both systems may be sensitive to PE, however, that does not necessitate that they are completely overlapping either and the authors should be careful to qualify these statements, as they can be read as going to the other extreme of arguing for complete overlap. The inverse relationship of signed PE with regions that typically show more activity during successful encoding into declarative memory does suggest that PE may engage the neural architecture of memory in a unique way that is not dependent on the type of memoranda or task characteristics (see additional comments on neural activity below), and distinct from how information is typically encoded into episodic memory.

We agree entirely with the reviewer's comments and have clarified this in the revised manuscript. Changes to the text are detailed in the related comment concerning neural activity below.

They also distance themselves from previous work on the hypercorrection effect by indicating that while their PE based explanation of learning is compatible with the attentional modulation proposed to strongly contribute to the hypercorrection effect, their findings reveal that inherent properties of PE are influencing learning directly. This sounds a bit like the Rescorla-Wagner vs. Pearce-Hall debate regarding the role of attention in learning theory (e.g., Roesch, Esber, Li, Daw, & Schoenbaum (2012) 10.1111/j.1460-9568.2011.07986.x). My recommendation is that if the authors wish to describe how their work is novel and differentiated from past theories on the PE effect in declarative memory, they need to present the past work on the hypercorrection effect in the introduction and describe a priori how their findings would differentiate a direct PE effect on learning from an effect mediated through attention. Simply giving a post hoc explanation and using italics for emphasis is not enough.

We thank the reviewer for the Rescorla_Wagner vs Pearce-Hall analogy and have introduced the hypercorrection effect in the introduction as suggested. To avoid an overly lengthy introduction we discuss in more detail the different interpretations of the hypercorrection effect and respective findings in the discussion. It is also important to note that we had no a-priori hypothesis regarding attentional versus PE effects on learning since this specific question was not what motivated the study. The broader fMRI aims of the study concern the implementation of a PE framework in declarative learning approached by exploring neural signals reflecting trial by trial parametric discrepancies in expectation-outcome informational mismatches, assessing whether they are valenced or absolute, and their relation to subsequent memory.

Accordingly, the revised introduction incorporates the hypercorrection effect and additional citations as follows:

“...Behaviourally, these episodes are sometimes remembered with greater fidelity in later memory tests^{22,31}. A related phenomenon, termed hypercorrection, is found in the error-correction field, whereby errors in tests of general knowledge are more likely to be corrected upon retest when they are committed with high confidence^{32,33}. This mismatch is associated with neural responses in cortical regions associated with error detection, surprise and attention³³⁻³⁵.”

The revised discussion elaborates on the different interpretations:

“The result that memory updating from feedback for erroneously answered questions, directly related to its PE magnitude, closely resembles the hypercorrection effect. The typical hypercorrection paradigm involves an initial test of general knowledge where confidence ratings are measured prior to feedback, followed by an immediate retest. Several studies show that high confidence incorrect answers are more likely to be corrected relative to low confidence incorrect answers³²⁻³⁵. This paradigm has been investigated with neuroimaging where it has been shown that high versus low confidence errors are associated with P3 like potentials in EEG³⁵ – similar to those elicited by ‘oddball’ stimuli – and ACC, DLPFC and TPJ activations in fMRI³⁴. The dominant explanation posited in this literature is that the surprise or ‘metacognitive mismatch’ arising from high confidence errors rallies attentional resources, resulting in enhanced learning. Our PE based explanation – supported by the powerful striatal correlates we observed – differs in that it reveals computations and properties inherent to the rules of declarative learning itself. These two approaches are reminiscent of the Rescorla-Wagner vs. Pearce-Hall debate, in which changes in associative strength are directly driven by PEs in the former and in the latter result from error based modulation of attention and event processing⁵¹ – but they are also compatible and may both provide explanatory utility here. For instance, the salience response we observed is consistent with the attentional explanation (see below). Procedural and methodological differences in the abovementioned hypercorrection fMRI study may account for why no striatal activation was observed. These include the aim and design of the fMRI analyses and behavioral measures, a significant temporal separation between the test, which was carried out prior to scanning, and the feedback (provided in the scanner), and the study-test-test design with novel material employed in the current study vs. the test-test general knowledge protocol.”

One of the most interesting aspects of the study was the finding that accurate memories of medium confidence (suggesting they have moderate memory strength) could be supplanted by false information (false feedback) more readily than weak memories. Although Fazio and colleagues (e.g., doi: 10.1177/0956797610371341) have shown the power of PE to correct high confidence false alarms, less work has been focused on the use of PE to instill misinformation (but see Fazio, Barber, Rajaram, Ornstein, & Marsh [2013]; doi: 10.1037/a0028649). Yet, this relatively novel component of the study is mentioned only in passing in the introduction.

We thank the reviewer for the suggestion and the link to the Fazio et al. work. We think there is value in keeping the introduction as brief as possible – outlying the overarching theme and background which motivated the study and is common to all of the individual results, namely PE based updating and whether it is applicable to declarative learning, without going into too much background about each facet of the results. We have made a revision to the introduction to better emphasize the novel application to false memory, and cite Fazio et al in the section of the discussion which deals with this result:

“...Thus, we were able to simultaneously assess the role of PEs in natural semantic learning and shed new light on the determinants of false memory formation.”

“...It has been suggested that weaker memories (e.g. old vs. new) are more amenable to manipulation⁵⁰, however one study showed that misinformation can be adopted even in place of general knowledge held with high confidence⁵¹.”

In addition, I had some difficulty following how false feedback was implemented in the study. The methods section indicates that 75% of trials with a confidence rating below 60 and 60% of trials with a confidence rating between 60-70 received false feedback, regardless of initial response accuracy. Responses with confidence above 70 (i.e., high confidence) are not given false feedback. Although the rationale for not giving false feedback for confidence ratings above 70 is given on pg. 8, the choice to generally give false feedback on such a large percentage of trials is not clear, particularly since the majority of lower confidence responses were errors. Their choice of bins for false feedback described in the methods section also makes translation of false feedback rates to the PE bins in the data analysis difficult; the PE bins for Test1 errors are in thirds (1-33, 34-66, 67-100) and the PE bins for Test1 corrects receiving false feedback are in still other bins (0-50, 50-70).

In the case of the correct-false analyses, the bins were chosen to make the number of responses in each bin equal, but this was not done for the analysis of information updating for incorrect responses (although there is a brief mention in the methods that if binned to have equal trials the effects were the same). In the end, it is difficult for the reader to get a good sense of how many (or the proportion) of the trials in each of the PE bins were false feedback vs. correct feedback.

Providing an analysis of “trial counts,” or at least proportions of trials with true and false feedback in each of the PE bins (perhaps in supplemental), would be helpful for understanding the effects of these binning choices, as well as with interpreting the data. Indeed, regarding these counts, if only ~30% of correct trials were entered into the false feedback analysis (presumably because most correct answers had high confidence), that is only ~10-11 trials total on average (based on there being on average 36 trials correct at Test1; if 64% of trials were incorrectly answered). If these were subdivided equally into “medium” and “low” bins, that means that only ~5 trials were in each of those bins, which is a very low number from which to derive analysis of memory updating.

The reviewer raises a number of points concerning the first (behavioural only) study, where we were interested in the possibility of supplanting veridical memories (correct answers) with false feedback (in addition to assessing the updating of knowledge for incorrectly answered questions). Here, we faced a challenge in that memory recall accuracy could only be determined post hoc (unlike recognition accuracy). Since we were unable to provide the false feedback solely for correctly answered questions, we had to

provide false feedback to both correct and incorrect answers – determined probabilistically with an algorithm. As the reviewer correctly points out, the majority of lower confidence responses were errors, and we set a limit on the high confidence end to avoid participants becoming aware of the manipulation. Therefore, since a) the overall proportion of correct answers was small (36%), and b) confidence in correct answers was skewed to the right (average 70%) (supplementary figures 1-2), the only way to generate a sufficient number of these (correct-false feedback low/medium PE) trials to enable the analysis was to provide a rather large proportion of false feedback answers in the low and medium confidence range. We weren't concerned by the large number of incorrect-false feedback trials this gave rise to, since we could treat these similar to incorrect-correct feedback trials for the purposes of the main (updating of incorrect trials) analyses, by scoring the trials on Test 2 relative to whatever feedback was presented in Test1 (whilst also checking that the incorrect-correct feedback trials alone gave rise to the main updating effects, and independently corroborating this in the second study, where there was no false-feedback condition). The reviewer is correct that despite the large number of false feedback trials, we were only able to capture around 5 low confidence and 5 medium confidence correct trials per subject in the false memory condition (i.e. average of 10 out of 36 correctly answered questions) and so this result should be viewed with caution and as a basis upon which to carry out a more extensive study.

Regarding the binning, we used two bins for the false memory analyses versus four bins in the incorrect updating analyses. The reasons for this were the smaller range of confidence values and smaller number of trials (as pointed out above) in the false memory condition. It was also important for this reason to split the trials into bins of roughly equal number, which we were able to achieve by grouping trials into bins of 1-49 and 50-70 (there were no 0 confidence correct answers and no false feedback trials above 70). Had we split the trials into bins based on an equal range of confidence values (e.g. 1-35, 36-70) there wouldn't have been a sufficient number of trials in the lower PE bin to perform a random effects statistical test.

Regarding the binning for the incorrect updating analysis, the larger number of trials enabled us to perform statistics based on bins of equal PE ranges (parametric), which is what we chose to present in the main results. However, as we mentioned in the methods section we also checked that the results held true for bins based on equal numbers of trials to make sure that the result was not affected by having more trials in the lower than upper negative PE bins (since incorrect answers tended to be expressed with lower confidence; Supplementary Figure 2). To test this, we used a non-parametric approach, ranking the negative PE values for each subject and placing an equal number of trials in each bin, from lowest to highest rank in thirds (and a separate bin for 0 PE trials). The downside of the non-parametric approach is that due to the aforementioned skew and tendency for participants to use round numbers in their confidence values, there were many trials of low negative PE values with the same rank which then had to be arbitrarily placed in adjacent bins to achieve equal trial numbers. Despite this 'working against' our hypothesis, the results of this test were still highly significant. Given the choice of presenting either method, we thought the parametric results were truer to the data and so chose that analysis, whilst also mentioning in the methods that we performed an additional test to control for uneven trial numbers in the bins, which yielded similar results.

In accordance with the reviewer's comments we have revised the methods section to better explain our rationale for the large proportion of false feedback trials in the recall study and difference in binning. We also added a cautionary note in the discussion regarding the small number of events in the false memory analysis. As requested, we have also provided the proportion of true vs. false feedback trials in each bin, in the Supplementary information (Supplementary Table 3):

"...The large proportion of false feedback trials was necessary because the veracity of recall answers could only be determined post testing, so we could not provide false feedback solely following correct answers. Furthermore, there were fewer correct answers (average 36%) and they tended to be expressed with higher confidence (confidence was above 70 – the cutoff for providing false feedback – in 56% of correct trials). Therefore, in order to generate sufficient trials where participants were supplied with false-feedback following correct answers of low and medium confidence, we resorted to high proportions within these confidence ranges – as a by-product, a large proportion of incorrect answers were met with false feedback (Supplementary Table 3)."

"...The reduction in number of bins reflected the smaller number of correct-false feedback trials (10/36 correct trials, on average) and the narrower PE range (limited at -70). For this reason, the trials were

split into two bins of roughly equal number to enable a random effects statistical analysis, which was achieved by creating PE bins of -1 to -49 and -49 to -70.”

“Note that the subsequent memory analysis in the recall study incorporated questions where participants skipped or erroneously answered the questions, and were supplied with false feedback (Supplementary Table 3). In these cases, accuracy of the Test2 answer was determined in accordance with the false feedback provided. When analyzing the learning of feedback following incorrect answers, we did not differentiate the two trial types (i.e. veridical and false feedback). Nevertheless, we performed an additional analysis excluding incorrect-false feedback trials. Indeed, the PE-subsequent memory relationship remained significant and the effect was even greater without these trials. We also ruled out the possibility that results were influenced by unequal numbers of trials in each bin by ranking the PEs and dividing them into three groups of equal number (as well as a group of 0 PE trials). Here too we observed a significant step-wise increase in accuracy as PE increased.”

“...However, we caution that our false memory result was based on a small number of trials and therefore warrants a more extensive study.”

“Supplementary Table 3. Proportion of erroneous answers according to PE bin and feedback type in the recall study.

Trial type as a percentage of the total number of incorrect answers in Test1. A significant number of false feedback answers were supplied in the recall study to determine false memory adoption following correct responses. Since erroneous answers entailed no/incorrect memory, we did not distinguish between false and original feedback when assessing memory updating, i.e., Test2 answers were scored with respect to the feedback supplied in Test1, not the original material. The updating effect for incorrect Test1 trials was also significant when excluding false-feedback trials from the analysis. “

Also, it was not always clear what “accuracy” meant when so many of the incorrect trials were also given false feedback. It is not accuracy in the objective sense, but in the sense of whether memory was updated with the false information, and thus, memory updating might be a better term. Notably, the “updating” term is used to only describe the dependent measure for the behavioral analysis of the fMRI study, but in that study false feedback was not given. In a related vein, I did not have a clear understanding from the main manuscript of how the updating metric for that study was calculated (Figure 2C) and why it was necessary to create this new metric, especially when the fMRI analyses themselves did not use this metric and instead used accuracy and confidence separately as DVs.

The reviewer correctly notes that with regards to updating from errors in the first (recall) study, ‘accuracy’ in Test 2 was determined with respect to the feedback in Test 1, whether the feedback was correct or false, so accuracy was objective with regards to the feedback and not the information in the original text (this was discussed in the methods section in the first draft). We now include a sentence in the results to emphasize this and avoid confusion.

“...In cases where incorrect answers/skipped questions were met with false feedback, accuracy in Test2 was scored with respect to the feedback, not the original text, such that an accurate updating of memory entailed providing the feedback answer (see methods)...”

The term updating is also our preferred description but is more general in the sense that it doesn’t specify which aspect of memory is being updated (content versus strength/confidence). We now use the term updating more evenly to describe the behavioural effects in both studies pointing out when it refers to content/accuracy or confidence/strength.

With regards to the combined behavioural (confidence and accuracy) updating measure that we describe and display in Fig 2C, the reviewer is correct that this was not strictly necessary with respect to our fMRI goals. However, for those in the memory field who have a greater interest in behaviour, it could be valuable in many instances to have a more holistic measure of memory updating performance using the combined metric, and since it is not immediately obvious as to how to achieve this, we thought it would be beneficial to delineate one solution, whilst also demonstrating the magnitude of the overall updating effect in our own behavioural data. Taking on board the reviewer's comments we have revised the section to better explain how the metric is calculated and what it represents as follows:

“The results demonstrate two forms of memory updating. The prediction error was not only related to the likelihood of successful learning and subsequent recall of the feedback information (memory content/accuracy), but also to the subjective strength of this learning (metamemory confidence/strength). We devised a method by which the two metrics could be combined to enable a holistic measure of memory updating. By summing the confidence level of each correct answer in Test2 (assigning 0 to those remaining incorrect) and dividing by the total number of incorrect answers for each unique Test1 PE value, we could calculate the overall updating effect and characterize its relationship to PE with linear regressions. In other words, an overall score of 100 would indicate that all incorrect Test1 trials for a given PE value were answered correctly and with a confidence of 100 in Test2. If only half the trials were answered correctly, and with an average confidence of 50, the overall updating value would be 25. In the recognition study (Fig. 2C), the group level regression shows that each unit increase in PE led to a 0.6% increase in this measure of updating ($\beta = 0.61$, intercept = 6.95, $R^2 = 0.85$, $p < 0.001$). Hence, overall updating from 0 PE feedback was 18.5% versus 80% from -100 PE feedback. This effect was also significant at the subject level (mean $\beta = 0.27 \pm 0.06$; $t_{(26)} = 4.13$, $p < 0.001$).”

Regarding the neural correlates of PE and subsequent memory, the finding of deactivation of DLPFC and hippocampus with negative prediction errors, despite these items being better encoded, would appear to be counter to most previous studies of successful encoding in declarative memory, including studies using reward and/or novel stimuli. Additionally, the inverse relationship of the hippocampus and PHC to salience is similarly counterintuitive, as if regions classically involved in declarative memory encoding are being inhibited during acquisition of information following negative PE. The authors need to address these findings in the context of past literature more directly, as well as the possibility of dissociations in neural mechanisms for PE based encoding and non-PE-based encoding in declarative memory.

We agree with the reviewer that negative PE deactivation related correlations with successful updating are counterintuitive in relation to the classic fMRI episodic encoding difference in memory (DM) findings, and this is an important point which we did give some attention to in the discussion. Due to constraints in manuscript length and the multidisciplinary scope of the paper, we perhaps did not discuss this in as much depth as necessary. As mentioned in the discussion, there are studies and meta-analyses which show that a number of regions exhibit a reverse DM effect, whereby deactivations during encoding correlate with subsequent memory accuracy – just as we found. Some of these regions are also nodes of the default mode network (DMN), which are known to deactivate in response to external stimuli and task engagement. Portions of the MTL are considered part of the DMN and likewise respond in a similar manner in many memory studies, by deactivating upon stimulus presentation relative to baseline. In fact, greater subsequent memory performance has also been shown to correlate with greater deactivations in MTL upon stimulus presentation. Hence, the profile of activity we see in these regions (inverse V response to the full range of PEs/saliency) is consistent with this literature – as is also noted by reviewer 3 (3rd comment) – because we would expect to see greater attention/task related effects as PE gets larger (positive or negative) and likewise greater updating. These are some possible explanations for the findings.

When considering how this study relates to traditional encoding DM paradigms, the most important point to consider is alluded to by the reviewer at the end of the comment and also in an earlier comment, namely, that the memory processes taking place here differ fundamentally from those in the aforementioned paradigm. In the typical study, encoding related activity corresponding with the presentation of simplistic, novel episodic stimuli (pictures/words) is determined in accordance with subsequent recognition performance for those stimuli. Here, we deal with the updating of complex semantic knowledge, which

differs from a simple episodic encoding task in three important ways. First, there was previous exposure to/encoding of the information in the study phase prior to scanning, hence when we perform a ‘remembered versus forgotten’ contrast during Test1, we aren’t looking at brain responses to information encountered for the first time in a manner akin to seeing a one shot novel picture stimulus which is subsequently recognized or not. Second and related, is that the feedback based updating often simultaneously entailed the alteration/overwriting of a competing memory as well as registration of a PE, rather than a pure formation of a novel memory trace. Finally, semantic and episodic memory acquisition is thought to depend on partially dissociable neural processes, with a greater cortical and basal ganglia involvement in the former, for instance, hippocampal lesion patients with anterograde episodic amnesia are able to acquire new semantic knowledge. For these reasons, we believe that attempting to view the updating related activations we observe in this paradigm through the lens of the classic DM literature is best avoided since we are dealing with different kinds. We agree with the reviewer that this distinction was not made well in the manuscript and have now revised and expanded the paragraph in the discussion concerning these various findings:

“Additionally, we found absolute PE, or saliency associated activations bilaterally in the DMPFC and anterior insula/IFG. These regions have been shown to signal saliency in several reward based imaging studies^{11,13,69}, respond to salient/surprising affectively neutral stimuli^{26,30,66}, and are associated with memory encoding^{81,82}. Of most interest in the saliency network were the numerous MTL clusters, primarily in the parahippocampal gyrus which is widely observed to be predictive of successful encoding⁸². Reward and PE related activity has been observed in PHC numerous times in fMRI studies⁶³, and on three occasions the response profile has been explicitly demonstrative of a salience, rather than a valenced PE signal^{69,83,84}. On two occasions, such activity was inversely correlated with outcome salience, as we observed^{84,85}. Moreover, enhanced PHC activity to unexpected or novel stimuli is frequently observed^{26,30,66}, and pharmacological agents have been utilized to show that parahippocampal repetition suppression effects to visual stimuli are modulated by dopamine⁸⁶, with effects on subsequent memory.

A primary function of saliency responses entails the marshalling of attentional resources to surprising/significant events. This is consistent with the increased activity in frontal regions and decreased activity in MTL, PCC and TPJ we observed⁸⁷. The latter are prominent nodes within the default mode network (DMN)⁸⁸, known to deactivate with external task engagement (relative to rest) and implicated in supporting episodic memory retrieval^{82,89}. This is thought to signify a shifting of attention from internal processes to external stimuli. Accordingly, the inverse V response we observed in these areas can be explained by greater attention/task related engagement as PE increases (positive or negative) and consequently greater updating.

The commonality in declarative memory encoding and PE/salience networks was further supported by a trial-by-trial subsequent memory analysis for feedback to erroneous answers. We found three cortical ROIs where feedback-related activity was predictive of Test2 veracity: right inferior parietal, right inferior frontal and posterior cingulate – in addition to the putamen ROI discussed above. Notably, in these regions, greater deactivations in response to feedback for incorrect answers were associated with greater Test2 accuracy. This finding is particularly interesting since PCC and right inferior parietal (TPJ) regions are notable for their *deactivation* as being associated with greater likelihood of subsequent memory for items^{82,90} (and are nodes in the DMN). These two regions are also significantly connected, anatomically and functionally to the parahippocampal gyrus (but not the HC) – where we also observed deactivations from baseline in response to increasing PE – leading to the suggestion that the PHG serves as the nexus through which the DMN interacts with the MTL memory system⁹¹. Indeed, the MTL – including hippocampus – is thought to be part of the DMN, and responds in a similar manner (deactivation in task vs. rest) in many memory studies^{87,89,92}. Subsequent memory performance has even been shown to correlate with greater deactivations in MTL at stimulus presentation⁹³. The inferior

parietal involvement in memory is a robust finding, this region is also sensitive to perceptual violations of expectancy, correlates with reward PEs and has been linked to violations of memory expectations^{94,95}.

These findings are *prima facie* at odds with many studies of declarative memory, where successful encoding is often associated with increased activity in frontal and MTL regions⁸². In the typical study, encoding related activity corresponding with the presentation of simplistic, novel episodic stimuli (pictures/words) is determined in accordance with subsequent recognition performance for those stimuli. In contrast to the aforementioned paradigm, the memory processes taking place here involve the updating of complex semantic knowledge, which differs in three fundamental ways. First, there was previous exposure to/encoding of the information in the study phase prior to scanning, hence a 'remembered versus forgotten' contrast for feedback in Test1 is not equivalent to looking at differential brain responses to information encountered for the first time, such as a one shot novel picture stimulus which is subsequently recognized or not. Second and related, is that the feedback based updating often simultaneously entailed the alteration/overwriting of a competing memory as well as registration of a PE, rather than the pure formation of a novel memory trace. Finally, semantic and episodic memory acquisition is thought to depend on partially dissociable neural processes¹⁴, with a greater cortical and basal ganglia involvement in the former⁴⁹. For instance, hippocampal lesion patients with anterograde episodic amnesia are able to acquire new semantic knowledge^{96,97}. For these reasons, attempting to view the updating related activations we observe in this paradigm through the lens of the classic memory encoding literature is best avoided."

It is possible that their use of PE ROIs for these analyses is overlooking other regions that show the more expected positive relationships between activity and salience and/or subsequent memory. A whole brain analysis of subsequent memory effects as a function of PE would be a helpful addition. Finally, given the proposed relationship between VS and MTL in salience/PE-based encoding, a model that examined the interactions between these regions (or used VS as a seed region to look at dynamic causal modeling) would help to provide more sophisticated insight into how the striatal PE is influencing regions classically associated with successful encoding in declarative memory.

The points discussed in the comment above regarding differences in traditional episodic memory subsequent memory studies, which also have a bearing on the possibilities suggested by the reviewer here. Regarding subsequent memory, if we understand the reviewer correctly, the suggestion would be to split the incorrect trials into e.g. high versus low PE bins in place of the parametric values, and then test for subsequent memory effects by PE interaction. This would however result in small trial numbers in each of the four conditions (in the region of 10) and low power. Also, there is not an obvious rationale to think that updated versus not updated trials in high versus low PE conditions would yield significant differences. With regards to regions outside the PE ROIs, carrying out a whole-brain analysis, as the reviewer suggested, did not yield any significant differences for subsequently updated versus not updated incorrect trials. We report this in the revised results.

"Carrying out a whole-brain analysis of incorrect trials (updated > not updated) did not reveal any regions that were predictive of content updating."

In regards to possible functional synchrony between VS and MTL, our findings counter the hypothesis that this would in fact be the case, since VS correlated with signed PE in a linear fashion, and MTL correlated inversely with unsigned PE. Therefore, it is implied that the interaction between the activation of these regions is not straightforward, and that the semantic updating process does not necessarily depend on such an interaction.

Modelling the causal relationships between the various PE related regions (network nodes) with DCM would be of significant interest but presents a number of challenges. The first concerns data acquisition. DCM is not recommended for EPI data acquired using interleaved slice acquisition. This problem is

possibly exacerbated since we used a 3 factor multiband interleaved sequence (1). A more general problem is that DCM is not recommended as a tool for exploratory analyses of data. This is because DCM is designed as a solution to a model comparison/selection problem, whereby Bayesian statistics are used to compute the likelihood of a number of equally plausible a priori hypotheses in a model space, given the data. Therefore, ideally a study should be designed with a paradigm that maximizes discriminability between pre-defined models and use DCM to determine which is most likely to have generated the observed data (model evidence) to select the optimal model (1,2). The HC-VS-VTA-HC loop described by Lisman and Grace offers one avenue which could be pursued and is often referred to in studies of reward/surprise based augmentation of recognition for single shot episodic stimuli. However, this specific mechanism is less likely to be involved in the semantic learning probed here – as we discussed at length in the points above. This may be why we did not observe any effects in the VTA, and why the MTL activation profile differed from the striatal activation profile.

1. Stephan et al. (2010). Ten simple rules for dynamic causal modelling. *Neuroimage*, 49:4, p3099-3109.
2. Daunizeau et al. (2011). Dynamic causal modelling: a critical review of the biophysical and statistical foundations. *Neuroimage*, 58:2, p312-22.

Minor issues:

Pg. 26: The authors indicate that most participants were entirely unfamiliar with the topic, but it is not clear how this determination was made. PE most likely becomes more important for learning in declarative memory when familiarity is low (see competing hypotheses for the hypercorrection effect). This may be particularly true for the apparent overwriting of accurate responses by false feedback, suggesting participants' inability to differentiate true and false information.

We chose an arcane topic, bearing in mind our target demographic of young Israeli students. The post-test debriefs confirmed that most participants were entirely unfamiliar with the topic prior to the experiment. There were some individuals who were aware of the most basic details relating to the Islands, i.e. that a war involving the UK and Argentina took place there, and their geographical location on the map. Beyond that, there was not a single participant who was aware of the history of their discovery, the various historical competing claims of sovereignty over the islands, their weather systems, wildlife and local geography, form of governance, main economic activity, detailed military tactics during the war, UN resolutions, place names, dates, distances, main actors and so on. To summarize, almost all of the information in the text was novel information for these participants. We also verified this by asking a number of volunteers to take the multiple choice (recognition) test used in the second experiment, without having read the text, to check that the demographic wouldn't be able to score above chance based on background familiarity. We have emphasised this in the revised manuscript accordingly:

“...We deliberately chose a topic that was arcane to our subject pool, and ensured that the information in the text was highly detailed and novel. Post-test debriefing confirmed that most participants were entirely unfamiliar with the topic. The recognition test was also piloted on a number of individuals who had not read the text, to verify that without study a score significantly greater than chance would be unattainable.”

Pg. 29: Why did the authors use mean replacement for missing data in all analyses except for the comparison of medium and low PE false feedback conditions?

The false-feedback comparison was carried out using a paired samples t-test (i.e. only 1 comparison between two conditions), whereas the other analyses relied on ANOVAs (F-test over four levels) followed by post-hoc comparisons. In the former case, participants with a missing value are essentially rendered useless since there is only one within-subject comparison, and the entirety of their data for one side of the comparison is missing. The three missing values also occurred in only one of the two conditions (the lower PE) and thus had a much larger impact than in the ANOVAs, both in terms of proportion of missing values and non-random distribution. On the other hand, removing a subject in the ANOVA with a missing value is

wasteful, since statistics can still be performed with the data in their other 3 levels and mean replacement has very little impact on the overall data set, whilst enabling those participants to be included in the ANOVA. We have included the following sentence in the revised manuscript to explain this difference:

“...excluding 3 subjects who did not have data in one of the two conditions (mean replacement being unsuitable for a within subjects test comparing only two conditions).”

Pg. 29: Should be “medium and low PE Test1 questions”? “To test for a difference in Test2 confidence of false feedback answers, between the subsets of high and medium PE Test1 questions answered corrected and supplied with false feedback (i.e., low PE, correct-to-false and medium PE, correct-to-false), we performed...”

We thank the reviewer for pointing out this error. Indeed, it should state medium and low PE Test1 questions, and we have rectified the mistake in the revised manuscript.

“To test for a difference in Test2 confidence of false feedback answers, between the subsets of low and medium PE Test1 questions answered correctly and supplied with false feedback (i.e. low PE, correct-to-false and medium PE, correct-to-false).”

Many of the references seemed to have a spurious ‘a’ peppered through the author names.

We thank the reviewer for pointing this out, we have removed them from the references.

Reviewer #3 (Remarks to the Author):

In this study, Pine et al. investigate whether the updating of semantic memories is determined by prediction errors, as has been suggested for other types of memory. This is an interesting question, and it is great seeing this information theoretic approach being applied to investigate how the brain deals with high-level, semantic information. The paper is well written and the methodology is sound. I have a few, mostly conceptual, comments and questions, which I detail below.

We thank the reviewer for these encouraging remarks, and hope that we have now satisfactorily addressed the constructive comments.

1. Negative prediction errors. As far as I am aware, negative prediction errors, in the reward literature, refer to the omission of an expected reward. However, in the current study, the author use this term to refer to the presentation of unexpected feedback. This does involve the omission of the expected feedback, but also the presentation of actual, be it unexpected, feedback. Therefore, I wonder if it might not be argued that this constitutes a positive prediction error. Could the authors comment on this?

When considering the applicability of RL models to this paradigm, we found that the most parsimonious way to think about the PE here is in terms of an outcome being better or worse than expected (based on the prediction) – this seems to be a fairly common definition within the RL literature. For instance, while the omission of an expected reward (or occurrence of an unexpected punishment) constitutes a negative PE, a reward of lower than expected magnitude also constitutes a negative PE. So here, we apply this by determining whether the information presented in the outcome confirms or negates what is expected or predicted by extant semantic knowledge (memory/belief), and to what degree. Therefore, in both cases where there are large PEs (low confidence correct answer [high positive PE], high confidence incorrect answer [high negative PE]) the feedback is unexpected, however this ‘unexpectedness’ alone does not confer a positive sign to the PE since the valence of the PE should be determined by whether the unexpected information negates or confirms a prior expectation/memory (and is therefore better or worse than expected). In line with the reviewer’s comments, we have expanded the relevant section of the results in the revised manuscript to express this more fully:

“...Key to this is being able to mathematically determine the PE (outcome – expectancy) for each trial of learning, which can be afforded by the confidence measure – a proxy of memory strength, and therefore of expectancy concerning the information conveyed by feedback. Based on the RL literature – where the magnitude and valence of a PE is equated with the degree to which an outcome is better or worse than expected – we posited that the semantic PE valence is determined by whether the feedback information confirms or negates what is expected or predicted by extant semantic knowledge, and its magnitude by the degree of strength (confidence) imparted in that prior knowledge/prediction. Accordingly, feedback to correct answers in Test1 stated with a confidence below 100 would elicit positive PEs (better than expected outcome) and vice versa for incorrect answers (where confidence above 0 would imply a worse than expected outcome). Thus, for incorrect answers $PE = -\text{confidence}$, and for correct answers $PE = 100 - \text{confidence}$. For example, an incorrect answer expressed with a high degree of confidence, e.g. 90% certainty that the answer was correct, would give rise to a large negative PE of -90, whereas an incorrect answer stated with only a 30% degree of certainty would evoke a PE of -30, upon encountering the feedback. Likewise, a correct answer stated with a low degree of confidence would be associated with a larger positive PE relative to one supplied with greater certainty.”

2. Neural correlates of signed prediction error. The authors interpret the PE responses as semantic PEs. However, could it be that participants perceive correct feedback as rewarding (and vice versa for negative feedback), and that the neural responses reported instead reflect signed reward prediction errors?

Please see our response to reviewer 1 (second comment) who also raised this important point.

3. Neural correlates of unsigned prediction error. Could this reflect an attentional effect, e.g. an alerting effect of surprising (salient) events? A reorienting of attention from internal to external signals, as a result of a surprising event, seems quite compatible with activation of frontal regions and deactivation of medial temporal regions.

The reviewer is absolutely right, we refer to this network as a ‘saliency’ network based on previous characterizations in the literature of regions exhibiting this response profile. A commonly expressed view about saliency responses is that one of their primary functions entails the marshalling of attentional resources to surprising/significant events, and this is indeed consistent with the activations we observe. As the reviewer points out, unsigned prediction errors were correlated with deactivations in various regions, including medial temporal structures, an effect that coincides with previously shown observations, whereby BOLD signal in the so called ‘default mode network’, including medial temporal lobes, decreases compared to resting epochs. Deactivations in these regions, elicited by engaging in various cognitive tasks, have been construed as signifying the shifting of attention from internal processes to external stimuli.

We emphasize the attentional component in the revised manuscript in the following passage of the results, and have cited some additional relevant papers:

“A primary function of saliency responses entails the marshalling of attentional resources to surprising/significant events. This is consistent with the increased activity in frontal regions and decreased activity in MTL, PCC and TPJ we observed⁸⁷. The latter are prominent nodes within the default mode network (DMN)⁸⁸, known to deactivate with external task engagement (relative to rest) and implicated in supporting episodic memory retrieval^{82,89}. This is thought to signify a shifting of attention from internal processes to external stimuli. Accordingly, the inverse V response we observed in these areas can be explained by greater attention/task related engagement as PE increases (positive or negative) and consequently greater updating.”

4. Before reading the neural correlates section, I was under the impression that confidence updating and accuracy updating were two sides of the same coin; they seemed to be treated such in the behavioural section. How do the authors interpret the fact that these two phenomena seem to correlate with distinct brain regions?

We were surprised to observe this neural dissociation for confidence and accuracy updating, but upon reviewing the (admittedly small) fMRI literature tackling the question of their individual neural bases, dissociation of their neural correlates appears to be a common finding (in episodic memory) both at encoding (for subsequent confidence and accuracy) as well as during retrieval (1-8, reviewed in 9). While there doesn't yet appear to be a consensus view on the distinct neural bases of each of these measures in this literature, subsequent confidence ratings often correlate with lateral PFC activity during encoding (i.e. neural activity during encoding correlated with subsequent confidence ratings), one of the two regions we observed here. It is also pointed out in this literature that at the behavioural level there are many instances where confidence and accuracy can be shown to diverge – these observations are a starting point for the suggestion that they may rely on distinct neural substrates. We have cited the most pertinent of these papers in the revised discussion accordingly:

“...We also observed a dissociation between regions correlating with subsequent confidence and accuracy. This finding is consistent with the observation that these two metrics can be shown to diverge behaviourally, and with fMRI studies that have reported dissociations in regions correlating with confidence and accuracy for episodic stimuli at encoding (i.e. subsequent memory), and during retrieval – suggesting that they rely on distinct neural substrates⁸⁸⁻⁹¹. Notably, subsequent confidence is often correlated with lateral PFC activity, one of the two major clusters we also observed to correlate with this measure.”

1. Chua et al. (2004). Dissociating Confidence and Accuracy: Functional Magnetic Resonance Imaging Shows Origins of the Subjective Memory Experience. *JOCN*, 16:7, p1131-42.

2. Qin et al. (2011). Subjective sense of memory strength and the objective amount of information accurately remembered are related to distinct neural correlates at encoding. *J Neurosci*, 31, p8920-7.

3. Richter et al. (2016). Distinct neural mechanisms underlie the success, precision, and vividness of episodic memory. *Elife*, 5: e18260.

4. Simons et al. (2010). Dissociation Between Memory Accuracy and Memory Confidence Following Bilateral Parietal Lesions. *Cereb Cortex*, 20(2):479-85.

5. Kim & Cabeza (2007). Trusting our memories: dissociating the neural correlates of confidence in veridical versus illusory memories. *J Neurosci*, 27 (45) 12190-12197.

6. Mendelsohn, Furman, & Dudai (2010). Signatures of memory: Brain coactivations during retrieval distinguish correct from incorrect memory. *Front. Behav. Neurosci.* 4, 1-12.

7. Moritz et al. (2006). Neural correlates of memory confidence. *Neuroimage*, 33(4):1188-93.

8. Chua et al. (2009). Neural correlates of metamemory: a comparison of feeling-of-knowing and retrospective confidence judgments. *JOCN*, 21; p1751-65.

9. Chua et al. (2014). The Cognitive Neuroscience of Metamemory Monitoring: Understanding Metamemory Processes, Subjective Levels Expressed, and Metacognitive Accuracy. *The Cognitive Neuroscience of Metacognition* City: Berlin, Heidelberg Publisher: Springer Berlin Heidelberg pp: 267-291.

5. For the discussion of declarative vs. non-declarative memory, in the introduction, the review paper by Henke (2010) that suggests that it may not be the type of memory (declarative vs. non-declarative, e.g.) that determines which brain regions are involved, but rather which type of neural computations are required for encoding that memory (Henke 2010), seems relevant.

We thank the reviewer for directing us to this relevant paper, which we cite in the revised introduction and discussion. The paper provides a valuable framework for how to view our results, particularly the approach by which semantic memory and associative learning can be married.

“...Along the same lines, the traditional view of a neurobiological, cortico-hippocampal, and midbrain-basal ganglia dissociation for declarative vs. non-declarative learning and memory, is being replaced by a more nuanced approach which favours interaction between memory systems^{33,41-43}. A PE based account of declarative learning would go a step further, by implying shared rules and neurobiological substrates between at least some forms of these seemingly disparate memories. The latter approach conforms to process based memory categorization, which distinguishes different forms of memory by the type of neural computation they depend on, rather than the involvement of consciousness⁴⁴...”

“...Thus, in process based accounts of memory systems⁴⁴, semantic memory shares with conditioning and procedural memory a reliance on slow encoding of rigid associations, subserved by basal ganglia, parahippocampal gyrus and neocortex.”

Reviewers' comments:

Reviewer #1 (Remarks to the Author):

Thanks for taking my comments so seriously and doing such an excellent job responding to the points I raised. The study is fantastic. Really really excellent. And I look forward to the follow ups you mention.

Reviewer #2 (Remarks to the Author):

I have reviewed the revised manuscript and am satisfied with the revisions in response to my original comments.

Reviewer #3 (Remarks to the Author):

In their revised manuscript, Pine et al. have addressed most of my concerns. I do however have some outstanding questions and remarks, which I detail below.

1. In the new analysis performed on the correct test 1 trials, described in response to Reviewer 1, could these results be explained by regression to the mean, or ceiling effects? That is, if participants were to respond randomly on both tests when reporting their confidence, one would also see a positive change for very low confidence scores on test 1, and negative changes for very high confidence scores, right? Along similar lines, a very low confidence score on test 1 has a lot of room to increase, whereas a high score cannot improve that much, because it's closer to the upper limit of the confidence scale.

2. I do not find the newly added discussion of whether the PE responses here reflect semantic or reward PEs (p. 26-27) very illuminating. The authors cite a number of studies with diverging findings, but it is not clear to me exactly what point they are trying to make.

3. Similarly, the new discussion on whether the updates are the result of the PEs themselves or the rallying of attention resources as a result of PE (p. 22), I find a bit vague: the authors seem to pose these as opposite camps in a debate, but also as compatible?

4. "Our PE based explanation – supported by the powerful striatal correlates we observed – differs in that it reveals computations and properties inherent to the rules of declarative learning itself." Am I right in understanding the authors to state that the involvement of the striatum shows that their effects are PE effects rather than attention effects? This would seem unwarranted reverse inference; why could attention not modulate the striatum? Rather, it seems important to deliver arguments for why the authors' paradigm would isolate PE effects, rather than attention effects, if such arguments are available.

Response to Reviewers

Reviewers' comments:

Reviewer #3 (Remarks to the Author):

In their revised manuscript, Pine et al. have addressed most of my concerns. I do however have some outstanding questions and remarks, which I detail below.

1. In the new analysis performed on the correct test 1 trials, described in response to Reviewer 1, could these results be explained by regression to the mean, or ceiling effects? That is, if participants were to respond randomly on both tests when reporting their confidence, one would also see a positive change for very low confidence scores on test 1, and negative changes for very high confidence scores, right? Along similar lines, a very low confidence score on test 1 has a lot of room to increase, whereas a high score cannot improve that much, because it's closer to the upper limit of the confidence scale.

We agree with the reviewer that regression to the mean or ceiling effects may contribute to the effect in this analysis. To help resolve this issue we have also included in the figure (Supplementary Fig. 4) the more straightforward relationship between Test1 and Test2 confidence, along with the best fit function that describes their linear relationship. What should be apparent from this figure is that we do not observe a classic ceiling effect relationship whereby the Y data increases linearly until it hits the ceiling and then remains flat at the ceiling level as X increases, such that the highest Y value is reached prior to the largest X value. For instance, if there were to be a uniform positive confidence shift from Test1 to Test2, taking the base value of 36 (the increase in confidence $[\Delta]$ at confidence₁=10 [PE=90]), we would have seen the ceiling hit at a PE of around 65 and remain at a constant level close to 100 as PE increases. Instead we observed a linear relationship with a slope of less than 1 (i.e. a steadily

decreasing shift), indicating a non-uniform increase in confidence, and no flat ceiling. With regards to regression to the mean based on random responses, our first thought is that this would imply a less plausible explanation for the effect, in that there would be no relationship between the confidence levels across the tests, as opposed to an explanation based on updating as a result of feedback provided in Test1. Even so, the data doesn't fit with the former hypothesis since purely random mean reversion would result in a flat line rather than a sloping line (we simulated this to check). Thus, at a PE of 50 (where there is equal room to increase and decrease – and therefore any such asymmetry would be factored out) we observed an increase in confidence of 20 points to a Test 2 level of 70. Furthermore, even a combination of a step shift with some mean regression wouldn't explain why as we move from the PE=50 point in equal steps to the left and right, the delta shift from the 20 point increase is not uniform (see delta-d50 row below).

conf1	10	20	30	40	50	60	70	80	90	100
conf2	46.3	53.0	56.4	60.2	69.8	69.3	73.4	83.4	85.2	92.7
delta	36.3	33	26.4	20.2	19.8	9.3	3.4	3.4	-4.8	-7.3
delta - d50	16.5	13.2	6.6	0.5	0	-10.4	-16.3	-16.4	-24.5	-27.1

In summary although this result may be partially accounted for by ceiling effects, it does not fully account for the relationship we observe. Without wanting to include this discussion in the manuscript, which would be distracting, we could also remove this result entirely if the reviewer thinks this would be optimal.

The revised supplementary figure 4 and legend, as well as results discussion have been accordingly changed:

“Finally, we also performed an analysis of confidence updating in trials answered correctly in both tests, to see whether there was a relationship between Test1 PE and change in confidence across tests. We observed a linear relationship between confidence in Test1 and confidence in Test2, with a slope of less than one (Supplementary Fig. 4; group level regression $\beta = 0.5$, intercept = 42, $R^2 = 0.98$), such that that the greater the PE in Test1, the greater was the confidence updating. Thus, for trials answered correctly with low confidence (large positive PE) a large increase in confidence was observed in Test2, whereas for higher confidence correct (low positive PE) trials, this metric gradually diminished and even became negative, reflecting a decrease in confidence in Test2. This result served as a validation of the PE hypothesis over an alternative hypothesis that the updating effects observed in the incorrect trials could be explained by low confidence being reflective of greater difficulty of learning that information. “

Supplementary Figure 4. Relationship between confidence updating and Test1 PE for trials answered correctly in both tests of the recognition study.

A. A significant relationship was observed between PE on Test1 for correctly answered questions (100-confidence) and confidence expressed in the same answers

in Test2. Notably, the slope of this relationship was significantly less than 1, indicating that as Test1 confidence increased (and positive PE decreased from 90 to 0), the degree of confidence updating decreased. Thus, questions answered correctly with low confidence in Test1 (high PE) were associated with large increases in confidence in Test2, whereas those answered with high confidence in Test1 (low PE) were associated with smaller increases, or even decreases in confidence in Test2 (despite having confirmatory feedback in Test1). Note, there were no Test1 trials answered correctly with zero confidence (equivalent to +100 PE). Displayed is group average confidence in Test2 (\pm SEM) for each confidence level in Test 1, along with the best fitting linear function. **B.** The delta, or change in average confidence from Test1 to Test2.

2. I do not find the newly added discussion of whether the PE responses here reflect semantic or reward PEs (p. 26-27) very illuminating. The authors cite a number of studies with diverging findings, but it is not clear to me exactly what point they are trying to make.

We have now modified the discussion regarding the semantic vs. reward responses to make our points clearer and more succinct. In short, we argue that performance related feedback activations in other tasks (in the absence of extrinsic reward) have not been observed to elicit reward and punishment signals in the VS in a way that could account for our results. Our literature review revealed that where striatal activity has been observed relating to cognitive performance outcomes, 1) positive and negative feedback both positively activated striatum, 2) the activation is often observed in DS, and 3) such activations are observed in tasks with a strong performance emphasis and/or trial and error learning over multiple rounds with the expectation of improvement, and/or with a stimulus-response component. We therefore concluded that there is not strong evidence for the idea that cognitive performance related feedback could act like a reward/punishment outcome for task performance and account for our VS findings. Rather, a component of the activity in the VS here likely relates to better or worse than expected outcomes in terms of informational mismatch (semantic PE), rather than solely to task performance signaling. The revised discussion now reads as follows:

“An important point to consider is whether the feedback linked VS PE activations in our task reflect the discrepancy between expected and observed information, as we hypothesise, or instead are indicative of reward (PE) responses associated with (expectation-outcome mismatches of) goal attainment. It is thought that purely cognitive feedback relating to task performance can be intrinsically motivating and engage the reward system. If this were the case, VS activations would be expected to exhibit reward and punishment responses in tasks that, like ours, do not involve extrinsic rewards. Probabilistic classification learning and visual categorization studies, where no extrinsic reinforcement is utilized and feedback following each trial only indicates whether the response was correct or incorrect, are illuminating in this regard. Such studies have yielded varying, inconclusive findings as to the response profile of VS to performance feedback⁷¹⁻⁷⁴. Commonly, both positive *and* negative feedback stimuli were found to increase VS activity, indicating that cognitive feedback

in itself (i.e. devoid of semantic content) is not akin to value based reward and punishment.

Feedback based caudate activity has been observed in declarative learning in the absence of extrinsic reinforcement, but in the DS. For example, in a paired associates word learning task involving multiple rounds, the caudate head was more activated by correct than incorrect feedback⁷⁵, although another caudate region (body) showed the reverse pattern, and in a subsequent study⁷⁶ positive and neutral feedback did not differ, whereas negative feedback was associated with a reduction in caudate activity and an increase in lentiform nucleus activity. Further studies utilising this paradigm^{67,75} showed that these differences also depend on the number of choice options and the stage of learning (i.e. which round).

In general, striatal responses to performance feedback are observed where motivation to perform is enhanced by an emphasis on task performance, either explicitly, or due to the nature of trial and error based paradigms where there is an expectation for improvement over trials. For example, differential feedback responses to positive and negative feedback on a card guessing task were observed in the DS (and not VS), but only when correct performance was financially rewarded⁷⁷. Additionally, the feedback typically comprises very simple cues which do not require deep semantic processing. Conversely, in our study the participants were told that they would likely answer many questions incorrectly, nor was there any ability to improve over trials, and the feedback was not overtly positive or negative but required deep semantic processing in order to evaluate the veracity of prior answers.

To summarize, the bivalent responses we observed in the VS are not accounted for by previous findings relating to performance related feedback, which do not resemble the standard reinforcement learning response evoked by extrinsic reward and punishment. Our findings therefore suggest that new information, as it relates to the unexpected confirmation or refutation of extant knowledge and beliefs, may be inherently rewarding or aversive, such that reinforcement is embedded in natural knowledge acquisition and updating via semantic PEs. However, it remains to be seen whether the VS would respond in the way we observed outside the context of a task – itself encoding a purely semantic PE, or instead combining semantic PE signals encoded elsewhere with explicit performance based reinforcement signals.”

3. Similarly, the new discussion on whether the updates are the result of the PEs themselves or the rallying of attention resources as a result of PE (p. 22), I find a bit vague: the authors seem to pose these as opposite camps in a debate, but also as compatible?

The cited paragraph concerns the similarities and differences between the paradigm we employed and our neuroimaging results to studies relating to the ‘hypercorrection’ effect, which is typically interpreted in terms of heightened attention to surprising feedback, which in turn benefits memory encoding. Importantly, if PE type updating is involved in semantic learning we would expect to observe regions which track the PE, which would differ in profile to an attentional response. Furthermore, we would expect to observe the involvement of striatal regions, which are not usually associated with attentional responses. What we therefore emphasized in this discussion section is that one of the unique and most robust findings in our study is the linear relationship between signed prediction errors (spanning from high negative to high positive PEs) and ventral striatum activation, implying the involvement of neural functions that are known to mediate PE based updating processes. We therefore posit that the involvement and correspondence of ventral striatum activation with semantic prediction errors is evidence in favor of a PE effect, meaning that an attention based explanation alone would not account for these findings. Nevertheless, attention is expected to be augmented during surprising events that characterize both high negative and positive PEs which is likely captured by the unsigned ‘salience responses’ depicted in Figure 4. What is furthermore interesting is that only in the signed PE regions were the activations predictive of subsequent memory, which suggests that PEs may be more important in mediating the updating effect than a generic salience/attentional response alone. In light of the reviewer’s comment, we have amended the relevant section of the paragraph which now reads as follows:

“...The dominant explanation posited in this literature is that the surprise or ‘metacognitive mismatch’ arising from high confidence errors rallies attentional resources, resulting in enhanced learning. Our PE based explanation differs in that it reveals computations and properties inherent to the rules of declarative learning *itself*. The VS response we observed is highly supportive of this approach, both because this region is not typically associated with attention, and also because the response profile linearly tracked the signed PE, whereas an attentional response would not differentiate surprising outcomes based on their sign. These two approaches are reminiscent of the Rescorla-Wagner vs. Pearce-Hall debate, in which changes in associative strength are directly driven by PEs in the former and in the latter result from error based modulation of attention and event processing⁵¹ – but they are also compatible and may both provide explanatory utility here. For instance, the salience response we observed is consistent with the attentional explanation (see below); notably, this network did not include striatal regions. Interestingly, only in the signed PE regions were the activations predictive of subsequent memory, which suggests that PEs may be more important in mediating the updating effect than a generic salience/attentional response alone...”

4. "Our PE based explanation – supported by the powerful striatal correlates we observed – differs in that it reveals computations and properties inherent to the rules of declarative learning *itself*." Am I right in understanding the authors to state that the involvement of the striatum shows that their effects are PE effects rather than attention effects? This would seem unwarranted reverse inference; why could attention not modulate the striatum? Rather, it seems important to deliver arguments for why the authors' paradigm would isolate PE effects, rather than attention effects, if such arguments are available.

We agree with the reviewer that the cited sentence may have failed to convey a clear message. As we state in response to comment 3 above, while attention effects are revealed and discussed in response to unsigned PEs (pp. 28-29), it is difficult to ascribe an attention-based explanation to the VS activation profile. This is because our results reveal a linear relationship between striatal activation and high

negative PE (expected to evoke surprise) to high positive PE (similarly expected to evoke surprise). Furthermore, VS is not typically associated with attentional responses. As detailed in the response above, we have modified this section to better convey these points.

REVIEWERS' COMMENTS:

Reviewer #3 (Remarks to the Author):

The authors have addressed all my concerns, and I have no further comments or questions.